# Dual Distillation for Few-Shot Anomaly Detection

**Le Dong[1], Qinzhong Tan[1,\*], Chunlei Li[2], Jingliang Hu[2], Yilei Shi[2], Weisheng Dong[1], Xiao Xiang Zhu[3], and Lichao Mou[2,†]**

[1]Xidian University, China
[2]MedAI Technology (Wuxi) Co. Ltd., China
 `lichao.mou@medimagingai.com`
[3]Technical University of Munich, Germany

## Abstract

Anomaly detection is a critical task in computer vision with profound implications for medical imaging, where identifying pathologies early can directly impact patient outcomes. While recent unsupervised anomaly detection approaches show promise, they require substantial normal training data and struggle to generalize across anatomical contexts. We introduce $D^2$4FAD, a novel dual distillation framework for few-shot anomaly detection that identifies anomalies in previously unseen tasks using only a small number of normal reference images. Our approach leverages a pre-trained encoder as a teacher network to extract multi-scale features from both support and query images, while a student decoder learns to distill knowledge from the teacher on query images and self-distill on support images. We further propose a learn-to-weight mechanism that dynamically assesses the reference value of each support image conditioned on the query, optimizing anomaly detection performance. To evaluate our method, we curate a comprehensive benchmark dataset comprising 13,084 images across four organs, four imaging modalities, and five disease categories. Extensive experiments demonstrate that $D^2$4FAD significantly outperforms existing approaches, establishing a new state-of-the-art in few-shot medical anomaly detection. Code is available at `https://github.com/ttttqz/D24FAD`.

## 1 Introduction

The automatic detection of anomalies in medical images is a crucial yet challenging task in clinical practice. Finding abnormalities early through automated screening enables timely intervention, leading to improved patient outcomes. However, developing robust algorithms for this task remains difficult due to the diverse manifestations of anatomical and pathological abnormalities in different patients. Furthermore, obtaining annotated datasets of verified anomalies is prohibitively expensive and time-consuming in medical contexts. These challenges have driven significant research interest in unsupervised anomaly detection methods, which can identify anomalies without requiring labeled abnormal training data.

Recent years have witnessed substantial advances in unsupervised anomaly detection approaches. Generative models, including autoencoders (Shvetsova et al., 2021) and generative adversarial networks (GANs) (Jiang et al., 2019), have shown promising results by learning to reconstruct normal patterns and identifying anomalies through reconstruction errors. Various architectural innovations have further improved detection performance, such as memory banks for storing and retrieving prototypical normal patterns (Gong et al., 2019), normalizing flows to model complex data distributions (Yu et al., 2021), and self-supervised learning for better feature representations (Li et al., 2021). Knowledge distillation has also emerged as an effective paradigm (Deng and Li, 2022),

---

⋆ Work done during an internship at MedAI Technology (Wuxi) Co. Ltd.
† Corresponding author.

where a student network learns to emulate a teacher's behavior on normal samples, allowing anomaly detection through discrepancy analysis.

These approaches, however, typically require large amounts of normal training data from target domains—data that may not be readily available in many clinical scenarios (Pachetti and Colantonio, 2024). In addition, they often struggle to generalize across different anomaly detection tasks, limiting their practical utility (Fernando et al., 2021; Cai et al., 2025). This highlights the need for few-shot anomaly detection, where models can adapt to new anatomical contexts using only a small number of normal reference images. Critically, these models are evaluated on novel tasks unseen during training to assess their ability to generalize across different anatomical structures and imaging conditions. In this paradigm, both training and inference are performed in an episodic manner, with each episode containing a support set of few normal images and a query image to be evaluated. This approach is particularly relevant in clinical practice, where physicians often have access to limited normal reference cases when examining new patients or encountering rare conditions. Despite its practical importance, few-shot anomaly detection in medical imaging remains relatively unexplored in the literature.

To address this important yet underexplored problem, we propose $D^24FAD$, a dual distillation framework for few-shot anomaly detection. Our approach leverages a pre-trained encoder as a teacher network to extract multi-scale feature maps from support and query images through parallel pathways with shared weights. A student decoder, trained exclusively on normal samples, learns to distill knowledge from the teacher network on query images while self-distilling on support images. This design enables anomaly detection at inference time by analyzing feature discrepancies between query and support representations, thereby using normal reference images as a basis for anomaly identification. Note that some works refer to this process as feature reconstruction. In this work, for consistency with prior anomaly detection methods (Salehi et al., 2021; Deng and Li, 2022; Gu et al., 2023; Ma et al., 2023; Zhou et al., 2022), we use the term distillation. Furthermore, we introduce a learn-to-weight mechanism that dynamically assigns importance to each support image, optimizing their reference values for different query images.

To facilitate research in this direction, we curate a comprehensive medical image dataset comprising four organs, four imaging modalities, and five disease categories, totaling 13,084 images. This dataset provides a standardized benchmark for evaluating few-shot anomaly detection methods across diverse medical contexts.

Our key contributions are:

- We formalize the few-shot medical anomaly detection task and establish a benchmark dataset spanning multiple organs, modalities, and pathologies.
- We propose a dual distillation framework that effectively leverages limited normal reference images through knowledge distillation for anomaly detection.
- We propose a learn-to-weight mechanism that evaluates the reference value of each support image conditioned on query images to enhance anomaly detection performance.
- We demonstrate through extensive experiments that our approach achieves state-of-the-art results, significantly outperforming existing methods.

## 2 METHODOLOGY

### 2.1 PROBLEM DEFINITION

We formulate few-shot medical anomaly detection as follows. Given a training set comprising several distinct tasks, each containing exclusively normal samples, we aim to develop an anomaly detection model with strong generalization capabilities for unseen anomaly detection tasks. During training, following the few-shot learning paradigm, we select one normal image as a query and several fixed normal images from the identical task as support samples to facilitate the learning of normal patterns. This task-agnostic model is not optimized for specific tasks but instead learns a universal anomaly detection strategy applicable across diverse medical images.

During inference, our test set contains both normal and abnormal images from novel target tasks unseen during training. We select $K$ normal samples from each target task to form a support set,

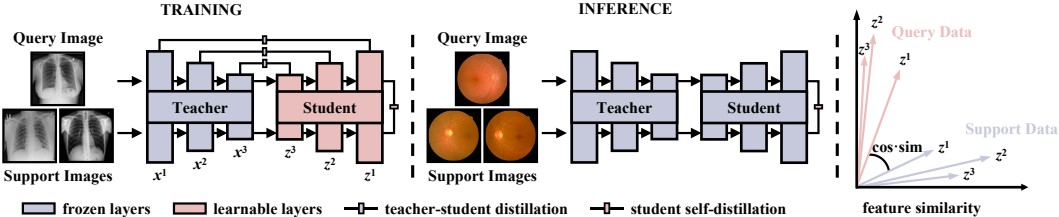

Figure 1: Overview of our dual distillation framework for few-shot anomaly detection. The architecture incorporates a frozen pre-trained teacher encoder and a learnable student decoder. During training, the teacher encoder processes both query and support images, while the student learns to reconstruct multi-scale feature representations through our proposed dual distillation approach. At inference time, for previously unseen tasks, anomalies are identified by analyzing discrepancies between query and support image features in the student network. In addition, we introduce a learn-to-weight mechanism that enhances model performance by dynamically assessing the reference value of each support image relative to a specific query (cf. Section 2.3).

where $K \in \{2, 4, 8\}$ in our experiments. This support set enables the model to adapt to previously unseen tasks when determining whether a test query image contains anomalies.

## 2.2 DUAL DISTILLATION

Figure 1 illustrates our proposed dual distillation framework. We prioritize architectural simplicity, with the network comprising only two components: a fixed pre-trained encoder and a learnable decoder. The encoder, functioning as a teacher, extracts visual features from both query and support images, while the decoder, acting as a student, reconstructs multi-scale feature representations from the encoder's output. We introduce two distillation strategies—teacher-student distillation and student self-distillation—that work in tandem for few-shot anomaly detection. The following subsections elaborate on these strategies.

**Teacher-Student Distillation** We leverage a pre-trained encoder as the teacher network, which has been exposed to diverse visual patterns through training on large-scale datasets such as ImageNet. This teacher encapsulates rich semantic knowledge that we transfer to guide our student decoder, providing a reference point for anomaly detection through consistency enforcement. We apply this distillation mechanism specifically to query images.

Formally, given a query image, let $\boldsymbol{x}^i_{\text{qry}}$ and $\boldsymbol{z}^i_{\text{qry}}$ denote feature maps from the $i$-th layer of the teacher encoder and student decoder, respectively. For knowledge transfer, we define the following loss function:

$$\mathcal{L}_{\text{tsd}} = \sum_i \frac{1}{H_i W_i} \sum_{h,w} 1 - \text{sim}(\boldsymbol{x}^i_{\text{qry}}[h,w], \boldsymbol{z}^i_{\text{qry}}[h,w]), \tag{1}$$

where $(h, w)$ indexes all spatial locations in feature maps $\boldsymbol{x}^i_{\text{qry}}$ and $\boldsymbol{z}^i_{\text{qry}}$, $H_i$ and $W_i$ represent the height and width of the feature maps at the $i$-th layer, and $\text{sim}(\boldsymbol{a}, \boldsymbol{b}) = \boldsymbol{a}^{\text{T}}\boldsymbol{b}/\|\boldsymbol{a}\|\,\|\boldsymbol{b}\|$.

After training exclusively on normal data, the teacher encoder and student decoder become aligned to represent normal patterns with high similarity. For normal test query samples that conform to patterns observed during training, the student's representations closely match the teacher's, resulting in low distillation loss. Conversely, when presented with anomalous query samples, the student decoder fails to adequately reconstruct the teacher's feature maps, producing discrepancies in their representations.

**Student Self-Distillation** Teacher-student distillation provides a solid foundation but is incomplete for few-shot anomaly detection since it does not make use of support images. Given limited normal references available in few-shot settings, we introduce student self-distillation to capitalize on these valuable normal examples. By performing self-distillation on support images, we establish a consistent reference base of what "normal" looks like, against which query images can be compared. This design aligns with clinical diagnostic practices, where distinguishing pathological abnormalities from normal anatomical variations requires reliable reference data.

Formally, given a support set, let $z_{\text{spt}}^{k,i}$ denote feature maps of the $k$-th support image from the $i$-th layer of the student encoder. We define a student self-distillation loss that aggregates multi-scale discrepancies as:

$$\mathcal{L}_{\text{ssd}} = \frac{1}{K} \sum_k \sum_i \frac{1}{H_i W_i} \sum_{h,w} 1 - \text{sim}(z_{\text{spt}}^{k,i}[h,w], z_{\text{qry}}^i[h,w]), \tag{2}$$

where $K$ represents the number of support images.

By minimizing this loss, the student network learns to align the query's representations with normal references, which is crucial for few-shot anomaly detection. This self-distillation mechanism enables the model to learn more discriminative features, even with the constraint of having access to only normal training samples.

**Training Objective**   In summary, we train our model by optimizing a combination of the two distillation losses:

$$\mathcal{L} = \lambda \mathcal{L}_{\text{tsd}} + \mathcal{L}_{\text{ssd}}, \tag{3}$$

where $\lambda$ is a hyperparameter that balances the teacher-student distillation and student self-distillation to the overall learning objective.

## 2.3   Learn-to-Weight

In Eq. (2), we initially assign equal weights to all support images. However, we observe that this assumption is often suboptimal—the reference value provided by each support image varies for different query images. This phenomenon is particularly prevalent in medical imaging, where individual anatomical variability leads to significant differences in shape and appearance among support images, resulting in varied reference values for the same query.

To better utilize the support set, we propose a learn-to-weight mechanism that adaptively weights support samples based on their relevance to a given query. Specifically, we learn reference values of support images conditioned on the query as follows:

$$w_i = \text{softmax}\left(\frac{z_{\text{qry}}^i \times \phi(z_{\text{spt}}^i)^{\text{T}}}{\sqrt{C_i}}\right), \tag{4}$$

where $\phi$ is a linear projection implemented as a $1\times1$ convolution. Note that appropriate reshaping operations are performed: $z_{\text{qry}}^i \in \mathbb{R}^{C_i \times H_i \times W_i}$ is reshaped to $\mathbb{R}^{1 \times (C_i \times H_i \times W_i)}$ and $z_{\text{spt}}^i \in \mathbb{R}^{K \times C_i \times H_i \times W_i}$ is reshaped to $\mathbb{R}^{K \times (C_i \times H_i \times W_i)}$.

We then reformulate the student self-distillation loss by incorporating our learn-to-weight mechanism as:

$$\mathcal{L}_{\text{ssd\_l2w}} = \sum_i \frac{1}{H_i W_i} \sum_{h,w} 1 - \text{sim}\left(\text{softmax}\left(\frac{z_{\text{qry}}^i \times \phi(z_{\text{spt}}^i)^{\text{T}}}{\sqrt{C_i}}\right) z_{\text{spt}}^i[h,w], z_{\text{qry}}^i[h,w]\right). \tag{5}$$

The overall training objective is updated to:

$$\mathcal{L} = \lambda \mathcal{L}_{\text{tsd}} + \mathcal{L}_{\text{ssd\_l2w}}. \tag{6}$$

Our learn-to-weight mechanism enables more effective knowledge transfer from support to query images by dynamically assessing the importance of each support sample based on its relevance to the specific query being evaluation.

## 2.4   Anomaly Scoring

At inference time, given a test query image and $K$ support images of a novel task (representing an unseen imaging modality and/or disease), we compute a similarity map at each scale using Eq. (5). These multi-scale similarity maps are then aggregated to derive a comprehensive anomaly map. The final anomaly score for the input image is obtained by computing the mean value of this aggregated anomaly map, providing a scalar quantification of anomalousness.

## 3 RELATED WORK

Given the expense and time required to obtain verified anomalies in medical imaging, unsupervised anomaly detection methods have become mainstream. Reconstruction-based approaches have emerged as a leading paradigm in this field. Schlegl et al. (2017) pioneer the use of GANs with AnoGAN (Li et al., 2018), later introducing f-AnoGAN (Schlegl et al., 2019), which improves efficiency by incorporating an encoder to map images to a latent space. Various autoencoder architectures have been explored, including variational autoencoders (Zimmerer et al., 2018) and vector-quantized variational autoencoders (Naval Marimont and Tarroni, 2021). To address the overgeneralization problem, where abnormal images are reconstructed too accurately, Gong et al. (2019) and Park et al. (2020) introduce memory banks to store normal patterns for comparison during inference. Several works (Rudolph et al., 2021; Gudovskiy et al., 2022; Yu et al., 2021) leverage normalizing flows, enabling exact likelihood estimation for image modeling, and achieve strong performance in anomaly detection.

Self-supervised learning (Jing and Tian, 2021) has also been applied to anomaly detection, typically following two paradigms. One-stage approaches train models to detect synthetic anomalies and directly apply them to real abnormalities (Tan et al., 2021; Schlüter et al., 2022). Two-stage approaches first learn self-supervised representations on normal data, followed by constructing one-class classifiers (Li et al., 2021; Sohn et al., 2021). Recently, knowledge distillation from pre-trained models presents another promising approach for unsupervised anomaly detection (Salehi et al., 2021; Deng and Li, 2022; Batzner et al., 2024). In these methods, a student network distilled by a pre-trained teacher network on normal samples can only extract normal features, leading to detectable discrepancies when anomalies are encountered during inference.

However, these methods typically require numerous normal samples for effective training—a requirement that becomes impractical in scenarios with limited data availability. This limitation motivates the need for more efficient anomaly detection methods that can achieve strong performance with few training samples.

Few-shot anomaly detection aims to generalize to novel tasks using limited reference samples. Early approaches such as TDG (Sheynin et al., 2021) and DiffNet (Rudolph et al., 2021) primarily focus on training with a small set of normal samples, adhering to a one-model-per-task paradigm that necessitates retraining for each new anomaly detection task. RegAD (Huang et al., 2022) addresses this limitation by leveraging registration as a task-agnostic proxy, enabling generalization across unseen tasks. It is noteworthy that these methods are primarily designed for industrial defects and fail to account for the high variability present in medical imaging, where data span diverse modalities (e.g., MRI and CT) and anatomical regions. In addition, their reliance on aggressive data augmentation (e.g., random rotation) is impractical for medical images with strict structural constraints.

In medical imaging, recent efforts like MediCLIP (Zhang et al., 2024) improve performance by synthesizing anomalies from limited normal data but remain confined to single-task scenarios. MVFA (Huang et al., 2024) introduces multi-level visual feature adapters to align CLIP's visual and language representations, yet demands pixel-level annotations and anomalous samples during training, which are costly to acquire in many scenarios. InCTRL (Zhu and Pang, 2024) achieves cross-domain generalization via contextual residual learning but still requires anomalous data for optimization. INP-Former (Luo et al., 2025) achieves reference-free anomaly detection by extracting intrinsic normal prototypes from test images. However, this approach is designed for natural images and shows limited performance on medical imaging data. In contrast, our method is trained solely on normal samples and, during inference, adapts to unseen anomaly detection tasks using only a few normal reference images, eliminating the need for annotated anomalies or retraining. This approach reduces deployment costs while maintaining strong generalization across medical modalities and anatomical regions.

## 4 EXPERIMENTS

### 4.1 EXPERIMENTAL SETTINGS

**Datasets**    To evaluate our method, we compile a comprehensive dataset spanning diverse anatomical regions, lesion types, and imaging modalities by integrating several widely used medical anomaly

Table 1: Anomaly detection performance comparison on multiple medical imaging datasets (HIS, LAG, APTOS, RSNA, and Brain Tumor). Performance is measured using image-level AUROC (%) with the best-performing method highlighted in **bold**. We compare our approach against both unsupervised (♠) and few-shot (♥) methods.

| Method | HIS | | | LAG | | | APTOS | | | RSNA | | | Brain Tumor | | |
|---|---|---|---|---|---|---|---|---|---|---|---|---|---|---|---|
| | 2-shot | 4-shot | 8-shot | 2-shot | 4-shot | 8-shot | 2-shot | 4-shot | 8-shot | 2-shot | 4-shot | 8-shot | 2-shot | 4-shot | 8-shot |
| UAE | $44.8_{\pm1.3}$ | $54.9_{\pm7.4}$ | $48.5_{\pm6.0}$ | $57.7_{\pm0.0}$ | $64.7_{\pm7.7}$ | $60.5_{\pm5.7}$ | $71.8_{\pm4.7}$ | $68.5_{\pm0.1}$ | $68.7_{\pm0.5}$ | $43.0_{\pm2.2}$ | $42.0_{\pm5.0}$ | $48.0_{\pm5.5}$ | $68.4_{\pm0.3}$ | $68.8_{\pm0.7}$ | $69.2_{\pm1.9}$ |
| f-AnoGAN | $67.4_{\pm2.3}$ | $49.5_{\pm11.1}$ | $54.0_{\pm9.5}$ | $51.1_{\pm3.6}$ | $55.0_{\pm2.9}$ | $37.1_{\pm8.1}$ | $35.5_{\pm8.0}$ | $31.3_{\pm5.2}$ | $30.5_{\pm3.7}$ | $38.6_{\pm6.2}$ | $43.1_{\pm1.2}$ | $42.2_{\pm2.6}$ | $34.7_{\pm5.3}$ | $53.9_{\pm4.5}$ | $65.3_{\pm2.0}$ |
| FastFlow | $73.1_{\pm1.4}$ | $75.1_{\pm0.7}$ | $71.2_{\pm1.4}$ | $66.3_{\pm2.8}$ | $66.5_{\pm2.1}$ | $72.2_{\pm1.5}$ | $87.8_{\pm1.4}$ | $90.0_{\pm0.9}$ | $84.1_{\pm1.3}$ | $74.4_{\pm0.7}$ | $77.9_{\pm0.7}$ | $79.6_{\pm0.5}$ | $90.8_{\pm0.4}$ | $90.3_{\pm1.1}$ | $91.0_{\pm0.6}$ |
| PatchCore | $67.8_{\pm0.1}$ | $64.7_{\pm0.5}$ | $59.6_{\pm0.9}$ | $69.1_{\pm0.6}$ | $67.5_{\pm1.1}$ | $77.1_{\pm0.7}$ | $67.3_{\pm1.4}$ | $69.1_{\pm1.3}$ | $71.9_{\pm0.6}$ | $65.0_{\pm0.2}$ | $70.9_{\pm0.3}$ | $74.6_{\pm0.1}$ | $64.1_{\pm1.9}$ | $74.0_{\pm2.3}$ | $78.1_{\pm1.6}$ |
| SimpleNet | $66.5_{\pm4.5}$ | $64.5_{\pm1.1}$ | $55.1_{\pm5.6}$ | $70.3_{\pm1.3}$ | $69.3_{\pm2.9}$ | $74.9_{\pm1.2}$ | $72.8_{\pm6.8}$ | $73.4_{\pm1.9}$ | $73.2_{\pm4.9}$ | $62.1_{\pm2.0}$ | $69.7_{\pm1.3}$ | $76.9_{\pm1.3}$ | $63.5_{\pm2.5}$ | $70.8_{\pm4.5}$ | $77.4_{\pm5.6}$ |
| ♠ CutPaste | $57.9_{\pm9.6}$ | $74.6_{\pm0.2}$ | $73.9_{\pm0.1}$ | $60.2_{\pm2.0}$ | $77.2_{\pm0.4}$ | $78.1_{\pm1.6}$ | $51.8_{\pm8.1}$ | $89.6_{\pm0.2}$ | $85.9_{\pm0.2}$ | $67.3_{\pm4.5}$ | $74.5_{\pm0.6}$ | $83.3_{\pm0.1}$ | $79.3_{\pm5.8}$ | $87.7_{\pm0.4}$ | $89.3_{\pm0.2}$ |
| NSA | $38.8_{\pm4.1}$ | $51.1_{\pm5.8}$ | $50.8_{\pm7.9}$ | $57.1_{\pm2.2}$ | $51.6_{\pm2.6}$ | $48.3_{\pm5.1}$ | $39.9_{\pm7.4}$ | $46.6_{\pm9.0}$ | $52.6_{\pm1.9}$ | $58.8_{\pm9.7}$ | $51.7_{\pm10.1}$ | $48.6_{\pm8.7}$ | $39.9_{\pm14.7}$ | $38.9_{\pm6.5}$ | $44.1_{\pm9.8}$ |
| ReContrast | $58.5_{\pm5.0}$ | $65.2_{\pm0.6}$ | $64.4_{\pm2.1}$ | $64.6_{\pm0.7}$ | $72.7_{\pm2.8}$ | $77.3_{\pm0.2}$ | $52.6_{\pm8.5}$ | $75.3_{\pm3.7}$ | $82.4_{\pm0.7}$ | $71.1_{\pm0.6}$ | $75.5_{\pm0.2}$ | $75.3_{\pm0.2}$ | $56.7_{\pm3.6}$ | $63.1_{\pm3.6}$ | $71.4_{\pm2.0}$ |
| RD++ | $39.6_{\pm4.6}$ | $40.3_{\pm3.5}$ | $41.8_{\pm5.5}$ | $42.7_{\pm2.3}$ | $41.9_{\pm2.6}$ | $43.5_{\pm2.0}$ | $54.6_{\pm6.8}$ | $47.0_{\pm7.5}$ | $52.4_{\pm5.7}$ | $59.1_{\pm1.5}$ | $60.0_{\pm2.2}$ | $58.5_{\pm2.2}$ | $26.0_{\pm4.9}$ | $25.5_{\pm3.3}$ | $24.4_{\pm3.4}$ |
| RD4AD | $73.2_{\pm1.0}$ | $68.1_{\pm0.2}$ | $68.1_{\pm0.7}$ | $65.7_{\pm2.0}$ | $71.7_{\pm1.3}$ | $77.0_{\pm0.9}$ | $51.3_{\pm4.4}$ | $68.1_{\pm4.7}$ | $67.4_{\pm2.0}$ | $60.7_{\pm1.7}$ | $68.2_{\pm1.4}$ | $76.1_{\pm0.7}$ | $62.2_{\pm6.0}$ | $74.6_{\pm1.9}$ | $81.4_{\pm1.2}$ |
| SCRD4AD | $68.6_{\pm1.9}$ | $69.5_{\pm1.0}$ | $69.5_{\pm2.2}$ | $68.1_{\pm1.4}$ | $67.1_{\pm0.3}$ | $68.8_{\pm1.9}$ | $82.9_{\pm4.7}$ | $84.9_{\pm3.3}$ | $83.6_{\pm1.7}$ | $67.7_{\pm0.7}$ | $67.9_{\pm1.0}$ | $67.3_{\pm0.4}$ | $76.5_{\pm4.8}$ | $81.8_{\pm1.0}$ | $81.4_{\pm1.7}$ |
| RegAD | $54.5_{\pm1.5}$ | $57.0_{\pm2.1}$ | $51.4_{\pm2.8}$ | $51.9_{\pm5.3}$ | $54.4_{\pm1.3}$ | $68.1_{\pm1.6}$ | $53.0_{\pm2.4}$ | $59.4_{\pm2.8}$ | $61.2_{\pm0.8}$ | $46.1_{\pm0.4}$ | $50.9_{\pm1.5}$ | $64.5_{\pm2.4}$ | $68.4_{\pm13.3}$ | $70.2_{\pm4.0}$ | $75.8_{\pm4.1}$ |
| WinCLIP | $58.0_{\pm0.0}$ | $58.8_{\pm0.0}$ | $57.2_{\pm0.0}$ | $55.0_{\pm0.0}$ | $53.8_{\pm0.0}$ | $55.0_{\pm0.0}$ | $53.1_{\pm0.0}$ | $53.1_{\pm0.0}$ | $53.0_{\pm0.0}$ | $69.6_{\pm0.0}$ | $70.8_{\pm0.0}$ | $71.8_{\pm0.0}$ | $38.2_{\pm0.0}$ | $45.0_{\pm0.0}$ | $43.4_{\pm0.0}$ |
| MediCLIP | $61.4_{\pm1.9}$ | $66.8_{\pm1.7}$ | $65.1_{\pm0.7}$ | $71.7_{\pm2.4}$ | $72.3_{\pm1.3}$ | $73.2_{\pm2.3}$ | $77.6_{\pm1.7}$ | $77.7_{\pm1.3}$ | $78.5_{\pm1.1}$ | $50.7_{\pm4.5}$ | $54.4_{\pm3.2}$ | $59.4_{\pm4.3}$ | $89.9_{\pm1.2}$ | $88.2_{\pm1.2}$ | $88.7_{\pm1.6}$ |
| ♥ MVFA-AD | $76.4_{\pm7.6}$ | $80.6_{\pm1.0}$ | $80.7_{\pm2.8}$ | $73.1_{\pm4.3}$ | $77.2_{\pm3.1}$ | $81.0_{\pm3.8}$ | $86.1_{\pm6.7}$ | $87.4_{\pm3.6}$ | $89.6_{\pm4.1}$ | $74.6_{\pm6.8}$ | $87.4_{\pm3.5}$ | $84.9_{\pm1.0}$ | $92.8_{\pm3.7}$ | $93.7_{\pm3.4}$ | $96.3_{\pm0.8}$ |
| InCTRL | $71.8_{\pm0.0}$ | $73.5_{\pm0.0}$ | $72.7_{\pm0.0}$ | $71.7_{\pm0.0}$ | $71.1_{\pm0.0}$ | $71.1_{\pm0.0}$ | $95.6_{\pm0.0}$ | $94.5_{\pm0.0}$ | $89.5_{\pm0.0}$ | $79.5_{\pm0.0}$ | $81.4_{\pm0.0}$ | $82.7_{\pm0.0}$ | $90.6_{\pm0.0}$ | $91.8_{\pm0.0}$ | $91.8_{\pm0.0}$ |
| INP-Former | $63.6_{\pm3.5}$ | $66.6_{\pm1.2}$ | $65.8_{\pm1.5}$ | $71.8_{\pm2.6}$ | $71.8_{\pm1.9}$ | $74.7_{\pm1.5}$ | $89.6_{\pm3.2}$ | $90.4_{\pm2.4}$ | $91.1_{\pm4.6}$ | $76.1_{\pm3.5}$ | $79.1_{\pm2.1}$ | $78.5_{\pm2.8}$ | $74.7_{\pm8.6}$ | $78.4_{\pm6.0}$ | $81.4_{\pm4.8}$ |
| AnomalyGPT | $47.9_{\pm1.1}$ | $47.1_{\pm0.5}$ | $48.3_{\pm0.4}$ | $57.6_{\pm1.3}$ | $58.1_{\pm0.8}$ | $57.5_{\pm1.0}$ | $76.2_{\pm0.7}$ | $78.7_{\pm0.4}$ | $77.6_{\pm1.1}$ | $62.8_{\pm0.4}$ | $63.0_{\pm1.0}$ | $62.7_{\pm0.6}$ | $76.9_{\pm3.1}$ | $78.9_{\pm2.4}$ | $78.9_{\pm2.7}$ |
| **Ours** | $\mathbf{94.2}_{\pm2.3}$ | $\mathbf{94.2}_{\pm3.3}$ | $\mathbf{94.3}_{\pm3.5}$ | $\mathbf{94.7}_{\pm2.2}$ | $\mathbf{96.2}_{\pm1.4}$ | $\mathbf{97.3}_{\pm0.8}$ | $\mathbf{100.0}_{\pm0.0}$ | $\mathbf{100.0}_{\pm0.0}$ | $\mathbf{100.0}_{\pm0.0}$ | $\mathbf{88.9}_{\pm10.4}$ | $\mathbf{97.9}_{\pm1.2}$ | $\mathbf{99.2}_{\pm0.5}$ | $\mathbf{95.5}_{\pm0.7}$ | $\mathbf{95.3}_{\pm1.0}$ | $95.5_{\pm0.8}$ |

detection benchmarks. We provide detailed information on dataset composition and train/test splits below to ensure transparency and reproducibility.

- **HIS:** Derived from Camelyon16 [*], which contains 6,091 normal and 997 abnormal patches from breast cancer patients, we allocate 700 normal images for training and create a balanced test set comprising 400 normal and 400 abnormal images.

- **LAG:** This dataset (Li et al., 2019) consists of 6,882 normal retinal fundus images and 4,878 abnormal images exhibiting glaucoma. Following Cai et al. (2022), we utilize 1,500 normal images for training and evaluate on 811 normal and 911 abnormal images.

- **APTOS:** This collection[†] of retinal images from diabetic retinopathy patients provides 1,000 normal samples for our training set, while the test set comprises 805 normal samples and 1,857 anomalous samples.

- **RSNA:** This chest X-ray dataset[‡] contains 8,851 normal and 6,012 lung opacity images. In accordance with Cai et al. (2022), we select 1,000 normal images for training and construct a balanced test set with 1,000 normal and 1,000 abnormal images.

- **Brain Tumor:** This dataset[§] encompasses 2,000 MRI slices without tumors, 1,621 with gliomas, and 1,645 with meningiomas. We classify both glioma and meningioma slices as anomalies. The normal cases originate from Br35H5 and Saleh et al. (2020), while the anomalous cases come from Saleh et al. (2020) and Cheng et al. (2015). Following Cai et al. (2022), our experimental protocol uses 1,000 normal slices for training and evaluates on a test set containing 600 normal and 600 abnormal slices (equally distributed between glioma and meningioma).

These datasets present significant challenges, including subtle pathological manifestations, heterogeneous lesion morphologies, and complex tissue architectures, thereby providing a rigorous testbed for evaluating our model's generalization capabilities across multiple medical imaging scenarios.

**Implementation Details** We employ WideResNet-50 (Zagoruyko and Komodakis, 2016) pre-trained on ImageNet as the backbone for our teacher encoder. Parameters are optimized using the Adam optimizer with $\beta_1 = 0.5$ and $\beta_2 = 0.999$ (Kingma and Ba, 2014). Our model is trained for 70 epochs with a batch size of 64, and all images are resized to $128 \times 128$ pixels. All experiments are conducted on a single NVIDIA A100 GPU.

---

[*] https://camelyon16.grand-challenge.org/Data/

[†] https://www.kaggle.com/c/aptos2019-blindness-detection

[‡] https://www.kaggle.com/c/rsna-pneumonia-detection-challenge

[§] https://www.kaggle.com/datasets/masoudnickparvar/brain-tumor-mri-dataset

Table 2: Ablation study results across five medical imaging datasets (HIS, LAG, APTOS, RSNA and Brain Tumor). We examine the impact of student self-distillation (`ssd`), teacher-student distillation (`tsd`), and learn-to-weight mechanism (`l2w`). Performance is reported as image-level AUROC (%) with the best configuration highlighted in **bold**.

| ssd | tsd | l2w | HIS | | | LAG | | | APTOS | | | RSNA | | | Brain Tumor | | |
|---|---|---|---|---|---|---|---|---|---|---|---|---|---|---|---|---|---|
| | | | 2-shot | 4-shot | 8-shot | 2-shot | 4-shot | 8-shot | 2-shot | 4-shot | 8-shot | 2-shot | 4-shot | 8-shot | 2-shot | 4-shot | 8-shot |
| - | ✓ | - | 66.1 | 66.7 | 74.1 | 64.4 | 70.2 | 75.9 | 58.2 | 60.8 | 63.9 | 57.0 | 66.5 | 74.1 | 55.5 | 73.8 | 81.1 |
| ✓ | - | - | 90.0 | 91.0 | 92.9 | 92.7 | 93.9 | 94.0 | 92.7 | 98.8 | 99.2 | 84.9 | 92.7 | 98.1 | 93.9 | 93.6 | 94.2 |
| ✓ | ✓ | - | 94.3 | 94.5 | 95.8 | 95.8 | 96.5 | 97.0 | **100.0** | **100.0** | **100.0** | 98.7 | 98.7 | 99.5 | 94.9 | 95.3 | 96.4 |
| ✓ | - | ✓ | 92.6 | 93.6 | 94.3 | 93.1 | 93.8 | **98.6** | 99.5 | 99.8 | 99.9 | 88.1 | 96.8 | 98.4 | 94.0 | 94.2 | 94.7 |
| ✓ | ✓ | ✓ | **96.7** | **98.4** | **99.0** | **97.6** | **97.9** | 98.0 | **100.0** | **100.0** | **100.0** | **99.0** | **99.1** | **99.8** | **95.1** | **96.3** | **96.8** |

**Competing Methods** Following prior works (Jeong et al., 2023; Huang et al., 2024; Gu et al., 2024), we compare our approach against both unsupervised and few-shot methods. We include a broad set of state-of-the-art methods, including the unsupervised approaches (UAE (Mao et al., 2020), f-AnoGAN (Schlegl et al., 2019), FastFlow (Yu et al., 2021), PatchCore (Roth et al., 2022), SimpleNet (Liu et al., 2023), CutPaste (Li et al., 2021), NSA (Schlüter et al., 2022), ReContrast (Guo et al., 2023), RD++ (Tien et al., 2023), RD4AD (Deng and Li, 2022) and SCRD4AD(Li et al., 2025)) and the few-shot methods (RegAD (Huang et al., 2022), WinCLIP (Jeong et al., 2023), MediCLIP (Zhang et al., 2024), MVFA (Huang et al., 2024), InCTRL (Zhu and Pang, 2024), INP-Former (Luo et al., 2025) and AnomalyGPT (Gu et al., 2024)). For unsupervised methods and few-shot methods without generalization capability (MediCLIP, MVFA, and INP-Former), we utilize $K$ support (normal) images as training data to train the models and evaluate them on the same task. For few-shot methods with generalization capability (RegAD, WinCLIP, InCTRL, AnomalyGPT, and ours), we adopt a **leave-one-out** training and testing protocol, where networks are trained on all datasets except the one being used for testing. Note that some baselines, such as PatchCore and RD4AD, compute image-level anomaly scores by applying max-pooling over the pixel-level anomaly maps.

**Evaluation Metrics** Following prior works (Cao et al., 2023; Pang et al., 2023; Zhu and Pang, 2024), we focus on image-level tasks, as many medical datasets do not provide pixel-level annotations. We use AUROC to quantify model performance, which is the standard metric for anomaly detection tasks.

## 4.2 Results and Discussion

We present our experimental results in Table 1. Across all evaluated settings ($K = 2, 4, 8$), our proposed method consistently outperforms competing approaches on the HIS, LAG, APTOS, and RSNA datasets, demonstrating significant improvements over the second-best methods. On the Brain Tumor dataset, our approach achieves performance comparable to MVFA. Unsupervised anomaly detection methods generally exhibit limited performance, likely due to insufficient training samples. In contrast, few-shot anomaly detection algorithms demonstrate superior performance, with our proposed approach showing particularly strong generalization abilities across diverse medical imaging datasets. With only two normal reference images ($K = 2$), our method achieves 100% precision on the APTOS dataset. Note that results for MediCLIP and InCTR differ from those reported in their original papers due to evaluation on different datasets.

For qualitative analysis, we demonstrate our method's anomaly localization capability through example anomaly maps in Appendix E.

In addition, we find that our model is not sensitive to the choice of support images (see Appendix C).

## 4.3 Ablation Studies

To evaluate each component's contribution to our approach, we conduct comprehensive ablation studies across five datasets, with results presented in Table 2.

**Student Self-Distillation** Student self-distillation plays a critical role in our few-shot anomaly detection framework. Without it, our network effectively degenerates into an unsupervised approach.

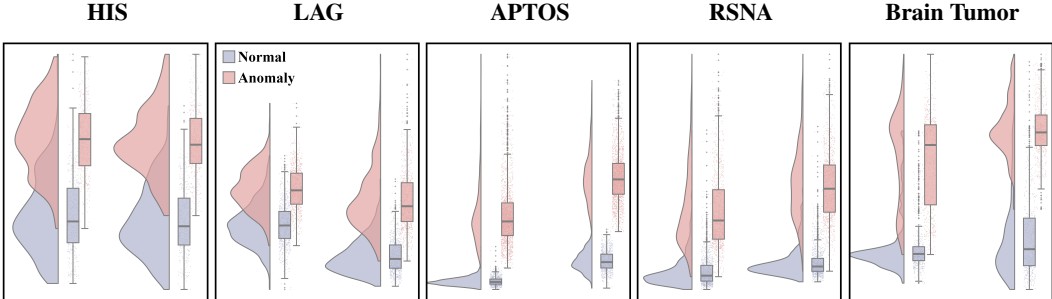

Figure 2: Distribution of abnormality scores for normal (blue) and abnormal (red) samples across five datasets, visualized using raincloud plots and boxplots. For each dataset, the left subplot presents results without the learn-to-weight mechanism, while the right subplot shows results with the mechanism applied. Reduced overlap between red and blue distributions indicates superior discrimination performance. Each boxplot displays the median value and interquartile range (IQR), with whiskers extending to the extrema within $1.5\times$ IQR from the quartiles, illustrating the significance of our approach.

Table 3: Analysis of weight coefficient $\lambda$ and its impact on anomaly detection performance.

| $\lambda$ | HIS | | | LAG | | | APTOS | | | RSNA | | | Brain Tumor | | |
|---|---|---|---|---|---|---|---|---|---|---|---|---|---|---|---|
| | 2-shot | 4-shot | 8-shot | 2-shot | 4-shot | 8-shot | 2-shot | 4-shot | 8-shot | 2-shot | 4-shot | 8-shot | 2-shot | 4-shot | 8-shot |
| 1.0 | 95.3 | 96.0 | 96.4 | 94.9 | 96.7 | 97.9 | **100.0** | **100.0** | **100.0** | 96.7 | 98.7 | 99.5 | 94.7 | 95.6 | 95.9 |
| 0.5 | 96.0 | 97.4 | 98.0 | 96.6 | **98.5** | 98.8 | **100.0** | **100.0** | **100.0** | 93.0 | **99.3** | 99.4 | 94.9 | 96.0 | 96.2 |
| 0.1 | **96.7** | **98.4** | **99.0** | **97.6** | 97.9 | 98.0 | **100.0** | **100.0** | **100.0** | **99.0** | 99.1 | **99.8** | **95.1** | **96.3** | **96.8** |
| 0.05 | 95.8 | 97.5 | 97.6 | 95.1 | 95.9 | 97.3 | **100.0** | **100.0** | **100.0** | 86.7 | 97.0 | 99.4 | 94.7 | 93.6 | 95.6 |
| 0.01 | 95.2 | 96.9 | 97.4 | 92.7 | 95.8 | **99.2** | 99.4 | **100.0** | **100.0** | 90.1 | 97.4 | 99.5 | 94.2 | 94.5 | 95.3 |

To rigorously validate its importance, we conduct an ablation study (see Table 2, first row). The results demonstrate a significant performance degradation when student self-distillation is removed, confirming its efficacy and necessity in the few-shot setting.

**Teacher-Student Distillation** Our results demonstrate that teacher-student distillation substantially enhances model performance. Under various few-shot settings ($K = 2, 4, 8$), AUROC scores across the five datasets improve by margins ranging from 0.66% to 13.76% (cf. Table 2). We include a qualitative t-SNE visualization in Appendix D to further examine the underlying mechanism.

**Learn-to-Weight Mechanism** Experimental results confirm the effectiveness of the proposed learn-to-weight mechanism, as evidenced in Figure 2. Without this mechanism, histograms of normal and abnormal samples exhibit significant overlaps. As detailed in Table 2, incorporating this component yields an average AUROC improvement of 1.91% across the five datasets for $K = 2, 4, 8$, with peak improvement reaching 6.81%. Moreover, this module is domain-agnostic and can be integrated into various few-shot learning frameworks.

**Weight Coefficient** The weight coefficient $\lambda$ balances the contribution between teacher-student distillation and student self-distillation in the overall objective function. To determine its optimal value, we conducted a grid search over $\lambda \in \{1.0, 0.5, 0.1, 0.05, 0.01\}$ using a dedicated validation set containing 100 normal and 100 anomalous images per task. Results in Table 3 indicate that setting $\lambda = 0.1$ yields the best average performance across all datasets on the test set.

## 4.4 IMPACT OF DIFFERENT BACKBONES

Table 4 presents a quantitative comparison of different backbones serving as the teacher network. As expected, deeper and wider networks demonstrate superior representational capacity, contributing to more accurate anomaly detection. Notably, our dual distillation approach maintains competitive performance even when employing more compact architectures (e.g., ResNet-34 (Koonce, 2021)). Interestingly, we observe that extremely large models such as Swin Transformer (Liu et al., 2021)

Table 4: Quantitative performance comparison of various backbone architectures employed as the teacher network across all datasets.

| Backbone | HIS | | | LAG | | | APTOS | | | RSNA | | | Brain Tumor | | |
|---|---|---|---|---|---|---|---|---|---|---|---|---|---|---|---|
| | 2-shot | 4-shot | 8-shot | 2-shot | 4-shot | 8-shot | 2-shot | 4-shot | 8-shot | 2-shot | 4-shot | 8-shot | 2-shot | 4-shot | 8-shot |
| ResNet-34 | 87.0 | 90.3 | 92.0 | 92.8 | 95.9 | 96.1 | 100.0 | 100.0 | 100.0 | 92.1 | 93.9 | 95.3 | 89.2 | 93.0 | 94.9 |
| ResNet-50 | 91.4 | 93.9 | 95.4 | 92.5 | 96.3 | 96.4 | 100.0 | 100.0 | 100.0 | 94.2 | 97.1 | 98.3 | 94.6 | 95.3 | 96.4 |
| Swin-S | 74.8 | 74.9 | 82.1 | 75.5 | 82.7 | 90.7 | 99.9 | 100.0 | 100.0 | 67.8 | 68.2 | 69.2 | 89.1 | 93.1 | 96.4 |
| Swin-B | 64.7 | 67.9 | 80.1 | 78.8 | 76.6 | 70.2 | 98.4 | 98.3 | 99.3 | 61.6 | 62.6 | 63.2 | 82.6 | 90.4 | 90.9 |
| CLIP | 88.5 | 91.2 | 93.5 | 90.3 | 93.1 | 93.8 | 97.8 | 99.9 | 99.9 | 85.7 | 90.4 | 93.1 | 92.8 | 93.5 | 95.2 |
| BiomedCLIP | 89.2 | 92.1 | 94.3 | 91.5 | 95.2 | 95.7 | 99.9 | 99.9 | 100.0 | 91.2 | 94.3 | 96.2 | 91.7 | 94.6 | **98.1** |
| WideResNet-50 | **96.7** | **98.4** | **99.0** | **97.6** | **97.9** | **98.0** | **100.0** | **100.0** | **100.0** | **99.0** | **99.1** | **99.8** | **95.1** | **96.3** | 96.8 |

yield suboptimal results despite their theoretical capacity advantage. This counter-intuitive finding suggests that excessive complexity in the teacher network may impede effective knowledge transfer to the student network, creating a representational gap that the distillation process struggles to bridge. The phenomenon highlights an important trade-off in our teacher-student architecture where the ideal teacher model balances representational power with distillation compatibility.

## 4.5 INFERENCE TIME AND PARAMETERS

We compare our model with competing few-shot anomaly detection methods and a representative unsupervised method (RD4AD) in terms of AU-ROC, inference time, and memory usage during inference, averaged across all five datasets (see Figure 3). All evaluations are conducted on a NVIDIA RTX 3090 GPU (24 GB VRAM). Our $D^2$4FAD achieves the highest AUROC score for anomaly detection while demonstrating significant computational efficiency—running $2\times$ faster than MVFA, $2.5\times$ faster than InCTRL and RegAD, and $3\times$ faster than AnomalyGPT and WinCLIP. In addition, $D^2$4FAD requires only 5 GB of memory for inference, positioning it among the most memory-efficient few-shot anomaly detection methods. These results demonstrate the superior practical value of our approach compared to competitors.

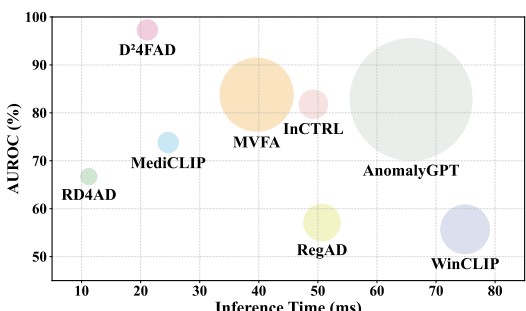

Figure 3: Performance comparison of anomaly detection methods across three dimensions: AUROC score (vertical axis), inference time (horizontal axis), and memory footprint (circle radius).

## 5 CONCLUSION

In this paper, we introduce $D^2$4FAD, a novel dual distillation framework for few-shot anomaly detection in medical imaging. By leveraging a pre-trained encoder as a teacher network and employing a student decoder that distills knowledge from the teacher on query images while self-distilling on support images, our approach effectively identifies anomalies in novel tasks using only a small set of normal reference images. The learn-to-weight mechanism we proposed further enhances performance by dynamically assessing the reference value of each support image conditioned on the query.

Our extensive experiments on a comprehensive benchmark dataset comprising 13,084 images across multiple organs, imaging modalities, and disease categories demonstrate that $D^2$4FAD significantly outperforms existing methods in few-shot medical anomaly detection. Specifically, our approach achieves superior AUROC scores while maintaining computational efficiency. Furthermore, $D^2$4FAD exhibits remarkable memory efficiency, requiring only 5 GB for inference, which positions it among the most resource-efficient few-shot anomaly detection methods.

The clinical significance of our work is particularly notable in medical settings where obtaining large annotated datasets is challenging due to privacy concerns, rare pathologies, or resource constraints. By requiring only a few normal samples, $D^2$4FAD enables practical anomaly detection across diverse medical imaging scenarios without extensive data collection efforts. This makes our approach

especially valuable for detecting anomalies in uncommon anatomical structures or rare disease presentations where comprehensive datasets are unavailable.

## ACKNOWLEDGMENTS

This work was supported in part by the Shaanxi Province Young Talent Support Program and the Qin Chuangyuan Innovation and Entrepreneurship Talent Project (H023360002), and in part by the China Postdoctoral Science Foundation (Nos. 2023M742737 and 2023TQ0257).

## ETHICS STATEMENT

The authors acknowledge that this work adheres to the ICLR Code of Ethics.

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

## APPENDIX

## A    USE OF LARGE LANGUAGE MODELS

Large language models were used solely for light editing tasks including grammar correction, spelling checks, and minor phrasing improvements to enhance clarity and concision.

## B    ANALYSIS OF PRE-TRAINING DATA FOR THE TEACHER NETWORK

To examine how the teacher's pre-training domain influences performance, we compare ResNet-50 models trained on ImageNet (natural images) and RadImageNet (Mei et al., 2022) (medical images). As shown in Table 5, the medically pre-trained teacher provides an average improvement of 0.91%, indicating that domain-aligned representations enhance accuracy. This also reinforces our main claim that the proposed distillation framework is flexible and can readily incorporate stronger or domain-specific teacher models.

## C    IMPACT OF SUPPORT IMAGE SELECTION

To evaluate the robustness of our method, we study the effect of support image choice. We perform five trials, each with randomly selected support examples, and report the mean and standard deviation. As shown in Table 6, performance with randomly selected support images differs only slightly from that with a fixed set, indicating that support sample selection has minimal impact on the results.

Table 5: Performance comparison between different pre-training datasets on ResNet-50 backbone.

|  | HIS | | | LAG | | | APTOS | | | RSNA | | | Brain Tumor | | |
|---|---|---|---|---|---|---|---|---|---|---|---|---|---|---|---|
|  | 2-shot | 4-shot | 8-shot | 2-shot | 4-shot | 8-shot | 2-shot | 4-shot | 8-shot | 2-shot | 4-shot | 8-shot | 2-shot | 4-shot | 8-shot |
| ImageNet | 91.4 | 93.9 | 95.4 | 92.5 | **96.3** | 96.4 | **100.0** | **100.0** | **100.0** | 94.2 | 97.1 | 98.3 | 94.6 | 95.3 | 96.4 |
| RadImageNet | **93.6** | **95.2** | **95.6** | **93.0** | 95.9 | **96.8** | 99.9 | **100.0** | **100.0** | **97.4** | **98.0** | **98.6** | **95.3** | **97.7** | **98.4** |

Table 6: Performance comparison between fixed (top) and randomly sampled (bottom) support sets.

|  | HIS | | | LAG | | | APTOS | | | RSNA | | | Brain Tumor | | |
|---|---|---|---|---|---|---|---|---|---|---|---|---|---|---|---|
|  | 2-shot | 4-shot | 8-shot | 2-shot | 4-shot | 8-shot | 2-shot | 4-shot | 8-shot | 2-shot | 4-shot | 8-shot | 2-shot | 4-shot | 8-shot |
| $D^2FAD_{fix}$ | $93.8_{\pm1.1}$ | $93.5_{\pm1.5}$ | $93.2_{\pm1.1}$ | $94.4_{\pm1.5}$ | $\mathbf{96.1}_{\pm1.3}$ | $\mathbf{97.1}_{\pm0.4}$ | $\mathbf{100.0}_{\pm0.1}$ | $\mathbf{100.0}_{\pm0.0}$ | $\mathbf{100.0}_{\pm0.0}$ | $88.4_{\pm9.7}$ | $\mathbf{97.3}_{\pm1.4}$ | $\mathbf{98.9}_{\pm0.4}$ | $\mathbf{95.8}_{\pm1.1}$ | $95.3_{\pm1.0}$ | $95.2_{\pm0.4}$ |
| $D^2FAD_{rnd}$ | $92.6_{\pm2.2}$ | $93.4_{\pm1.1}$ | $\mathbf{93.5}_{\pm0.9}$ | $\mathbf{94.8}_{\pm1.9}$ | $95.8_{\pm1.3}$ | $96.5_{\pm0.8}$ | $99.8_{\pm0.5}$ | $\mathbf{100.0}_{\pm0.0}$ | $\mathbf{100.0}_{\pm0.0}$ | $\mathbf{93.4}_{\pm6.9}$ | $96.9_{\pm2.1}$ | $98.4_{\pm0.9}$ | $94.7_{\pm0.5}$ | $\mathbf{96.0}_{\pm1.3}$ | $\mathbf{96.1}_{\pm0.6}$ |

## D  TEACHER-STUDENT DISTILLATION

We visualize feature representations from the student network using t-SNE on the RSNA dataset. As illustrated in Figure 4, using teacher student distillation produces more compact and discriminative feature clusters, validating its effectiveness.

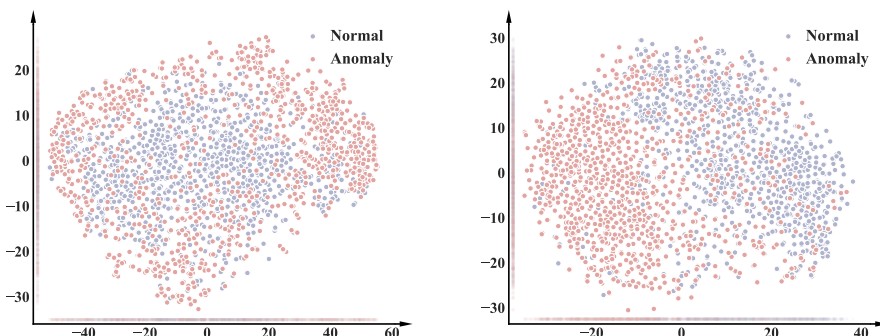

Figure 4: t-SNE visualization of embeddings from normal and abnormal samples in the RSNA dataset, extracted from the student network. Left: embeddings without teacher-student distillation. Right: embeddings with teacher-student distillation applied.

## E  QUALITATIVE RESULTS

For qualitative analysis, we demonstrate our method's anomaly localization capability using anomaly maps shown in Figure 5. The figure displays results across five datasets: HIS, LAG, APTOS, RSNA, and Brain Tumor (from left to right). The top row shows original images from each dataset, while the bottom row shows the corresponding anomaly maps generated by our method. Despite promising qualitative localization, the absence of pixel-level annotations in our benchmark hinders quantitative evaluation, and this limitation could be addressed with fully annotated datasets in the future.

## F  ALTERNATIVE INSTANTIATIONS OF THE LEARN-TO-WEIGHT MECHANISM

To assess the flexibility of our weighting mechanism, we introduce three alternative similarity functions—Gaussian, embedded Gaussian, and concatenation—to compute query–support similarity, followed by softmax normalization:

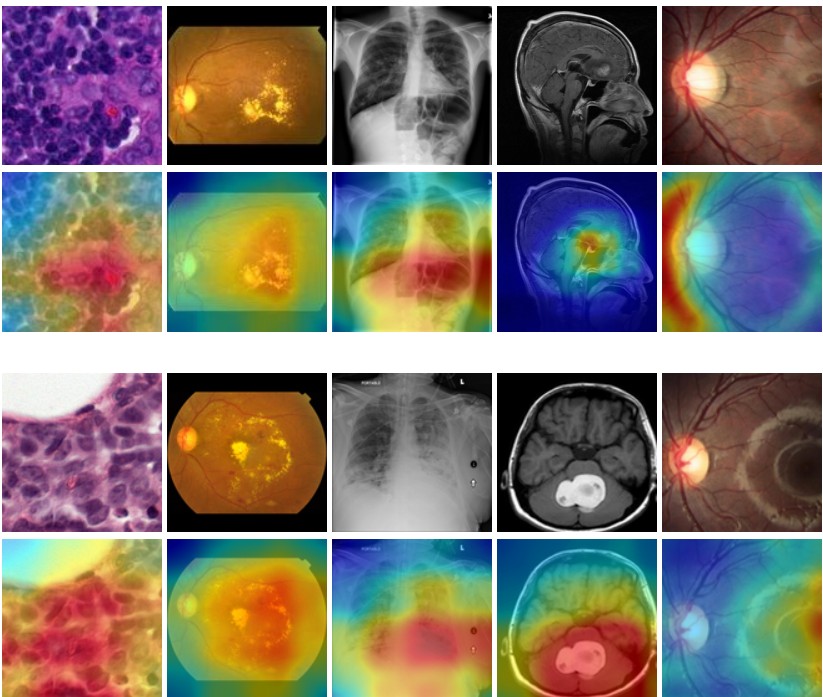

Figure 5: Visualization of exemplar anomaly maps generated by the proposed model.

**Gaussian**

$$\boldsymbol{w}_i = \text{softmax}(e^{\boldsymbol{z}_{\text{qry}}^i (\boldsymbol{z}_{\text{spt}}^i)^{\text{T}}}). \tag{7}$$

Euclidean distance, as in (Buades et al., 2005), is also applicable, but the dot product is more implementation-friendly in modern deep-learning frameworks.

**Embedded Gaussian**

$$\boldsymbol{w}_i = \text{softmax}(e^{\theta(\boldsymbol{z}_{\text{qry}}^i)\phi(\boldsymbol{z}_{\text{spt}}^i)^{\text{T}}}), \tag{8}$$

where $\theta$ and $\phi$ are linear projections implemented with $1 \times 1$ convolutions.

**Concatenation**

$$\boldsymbol{w}_i = \text{softmax}(\text{ReLU}(\boldsymbol{w}^{\text{T}}[\theta(\boldsymbol{z}_{\text{qry}}^i), \phi(\boldsymbol{z}_{\text{spt}}^i)])), \tag{9}$$

where $[\cdot, \cdot]$ denotes concatenation, and $\boldsymbol{w}$ is a learnable weight vector. The query feature $\boldsymbol{z}_{\text{qry}}^i \in \mathbb{R}^{1 \times C_i \times H_i \times W_i}$ is broadcast to $\mathbb{R}^{K \times (C_i \times H_i \times W_i)}$ for this computation.

Table 7: Performance comparison between different ways to instantiate Learn-To-Weight.

| | HIS | | | LAG | | | APTOS | | | RSNA | | | Brain Tumor | | |
|---|---|---|---|---|---|---|---|---|---|---|---|---|---|---|---|
| | 2-shot | 4-shot | 8-shot | 2-shot | 4-shot | 8-shot | 2-shot | 4-shot | 8-shot | 2-shot | 4-shot | 8-shot | 2-shot | 4-shot | 8-shot |
| Gaussian | 95.7 | 97.4 | 98.0 | 95.6 | 96.9 | 97.0 | 99.9 | 99.2 | **100.0** | 97.0 | 98.1 | 98.8 | 94.1 | 95.3 | 95.8 |
| Gaussian, embed | 96.0 | **98.6** | 98.4 | 96.9 | 97.2 | 97.4 | 99.9 | **100.0** | **100.0** | 98.8 | 98.7 | 99.2 | 94.5 | 95.7 | 96.2 |
| Concatenation | 95.9 | 97.6 | 98.2 | 96.7 | 97.1 | 97.7 | 99.9 | 99.8 | **100.0** | 97.5 | 97.9 | 99.0 | 94.3 | 94.5 | 96.0 |
| Ours | **96.7** | 98.4 | **99.0** | **97.6** | **97.9** | **98.0** | **100.0** | **100.0** | **100.0** | **99.0** | **99.1** | **99.8** | **95.1** | **96.3** | **96.8** |

The remaining steps follow Section 2.3. As shown in Table 7, different weighting strategies produce comparable but distinct results. The scaled dot-product formulation used in our main model shows slightly better performance, likely because it provides a more direct and effective measure of feature similarity. These variants demonstrate the flexibility of our learn-to-weight design, and other alternatives may further improve performance.

We also visualize the learned weights in Figure 6. Support images that are normal, anomalous, or semantically irrelevant are marked in green, red, and yellow, respectively. Across multiple layers

(weight 1/2/3), the model consistently assigns lower weights to anomalous or irrelevant support images, highlighting the effectiveness of the proposed mechanism.

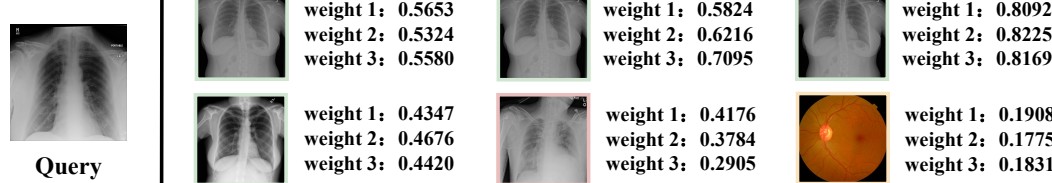

Figure 6: Visualization of learned weights in the learn-to-weight mechanism.

