# OpenReview forum: "Dual Distillation for Few-Shot Anomaly Detection"
_ICLR.cc/2026/Conference — ICLR 2026 Poster_

### Official Review · Reviewer_9qcV · 2025-10-20

**Soundness:** 3
**Presentation:** 2
**Contribution:** 3
**Rating:** 4
**Confidence:** 4

**Summary:**

This paper focuses on image-level anomaly detection across anatomical contexts. It proposes a dual distill framework for anomaly detection in few-shot setting, where only up to 8 normal images as references. Specifically,  a spatial feature reconstruction loss is implemented for query images, and a spatially discrepancy loss between the query image and references images are implemented. The proposed framework is highly closed to the realistic medical setting, the proposed framework achieves satisfactory results.

**Strengths:**

Strengths:
- This paper is well-written and easy to follow.
- The proposed dual distillation framework is effective and easy to implement.
- The experiments are extensive.

**Weaknesses:**

Weaknesses:
 -  The introduction part from Line 52- Line 54 can be better motivated.
   - 1] actually leverages a large amount of unlabeled abnormal data for anomaly detection. I don’t quite understand why the authors cite this paper here, since they are arguing that, in some cases, normal data can be rarer than abnormal data.
   -  The authors should have supported reference papers for the statement/claim from Line 54-55.
- Figure 1 is not very informative. For example, the authors could include notations such as  X to represent the feature space of the teacher encoder and Z for the space of the student decoder. In addition, an illustration showing how the proposed loss terms are computed would make the figure more informative and easier to understand.
 - The title and the abstract are a bit mis-leading. After reading the paper, it seems to me that the inference is few-shot but not the training. My understanding is that only $K$ images for training if the setting is $K$-shot. However, the Line 291-309 suggest the training requiring the whole dataset. Please clarify this if I am wrong.

Minor weaknesses:
- Line 63 should be student decoder.

**Questions:**

See weaknesses.

---

> ### Author Response · Authors · 2025-11-27
>
> >The introduction part from Line 52- Line 54 can be better motivated.
>
> Many thanks for the suggestion. We have revised this part to improve the motivation in the updated version.
>
> “*These approaches, however, typically require large amounts of normal training data from target domains---data that may not be readily available in many clinical scenarios [1]. In addition, they often struggle to generalize across different anomaly detection tasks, limiting their practical utility [2,3]. This highlights the need for few-shot anomaly detection, where models can adapt to new anatomical contexts using only a small number of normal reference images. Critically, these models are evaluated on novel tasks unseen during training to assess their ability to generalize across different anatomical structures and imaging conditions. In this paradigm, both training and inference are performed in an episodic manner, with each episode containing a support set of few normal images and a query image to be evaluated. This approach is particularly relevant in clinical practice, where physicians often have access to limited normal reference cases when examining new patients or encountering rare conditions. Despite its practical importance, few-shot anomaly detection in medical imaging remains relatively unexplored in the literature.*”
>
> [1]Eva Pachetti and Sara Colantonio. A systematic review of few-shot learning in medical imaging. Artificial Intelligence in Medicine, 156:102949, 2024.
>
> [2]Tharindu Fernando, Harshala Gammulle, Simon Denman, Sridha Sridharan, and Clinton Fookes. Deep learning for medical anomaly detection - A survey. ACM Computing Surveys, 54(7):141, 2021.
>
> [3]Yu Cai, Weiwen Zhang, Hao Chen, and Kwang-Ting Cheng. MedIAnomaly: A comparative study of anomaly detection in medical images. Medical Image Analysis, 102:103500, 2025.

---

> ### Author Response · Authors · 2025-11-27
>
> >1] actually leverages a large amount of unlabeled abnormal data for anomaly detection. I don’t quite understand why the authors cite this paper here, since they are arguing that, in some cases, normal data can be rarer than abnormal data.
>
> We apologize for the lack of clarity. Our original intent was to cite this reference to support the statement that “verifying them as ‘truly normal’ still requires expert review and institutional approval.” In the revised version, this citation has been removed because we reorganized the paragraph to better illustrate the motivation.

---

> ### Author Response · Authors · 2025-11-27
>
> >The authors should have supported reference papers for the statement/claim from Line 54-55.
>
> We thank the reviewer for pointing this out. As suggested, we have added the following two references to support the statement.
>
> [1]Tharindu Fernando, Harshala Gammulle, Simon Denman, Sridha Sridharan, and Clinton Fookes. Deep learning for medical anomaly detection - A survey. ACM Computing Surveys, 54(7):141, 2021.
>
> [2]Yu Cai, Weiwen Zhang, Hao Chen, and Kwang-Ting Cheng. MedIAnomaly: A comparative study of anomaly detection in medical images. Medical Image Analysis, 102:103500, 2025.

---

> ### Author Response · Authors · 2025-11-27
>
> >Figure 1 is not very informative. For example, the authors could include notations such as X to represent the feature space of the teacher encoder and Z for the space of the student decoder. In addition, an illustration showing how the proposed loss terms are computed would make the figure more informative and easier to understand.
>
> Thanks a lot for the suggestion. We have updated Figure 1 accordingly.

---

> ### Author Response · Authors · 2025-11-27
>
> >The title and the abstract are a bit mis-leading. After reading the paper, it seems to me that the inference is few-shot but not the training. My understanding is that only K images for training if the setting is K-shot. However, the Line 291-309 suggest the training requiring the whole dataset. Please clarify this if I am wrong.
>
> Our approach follows the **episodic** training paradigm widely used in meta-learning [1,2], which involves two phases:
>
> Meta-training: The entire training dataset is used to construct many simulated few-shot tasks. Each task includes a support set (with K normal samples) and a query set. By training across these episodes, the model learns to adapt quickly to new tasks using only limited samples.
>
> Testing: During evaluation, we strictly follow the K-shot setting. For each test task, the model adapts using only the provided K normal samples, with no additional data or access to anomalous examples.
>
> We believe the confusion stems from differences between standard supervised training and meta-learning conventions. We have revised the manuscript to more clearly explain the episodic training mechanism and avoid potential misunderstandings.
>
> [1]Oriol Vinyals, Charles Blundell, Timothy Lillicrap, Koray Kavukcuoglu, and Daan Wierstra. Matching networks for one shot learning. In Advances in Neural Information Processing Systems (NeurIPS), pages 3630-3638, 2016.
>
> [2]Chelsea Finn, Pieter Abbeel, and Sergey Levine. Model-agnostic meta-learning for fast adaptation of deep networks. In Proceedings of the International Conference on Machine Learning (ICML), pages 1126–1135, 2017.

---

### Official Review · Reviewer_2kEb · 2025-10-27

**Soundness:** 3
**Presentation:** 3
**Contribution:** 3
**Rating:** 6
**Confidence:** 3

**Summary:**

This paper introduces D²FAD, a novel framework for few-shot anomaly detection (FAD), with a specific focus on medical imaging. The core problem addressed is the scarcity of normal reference images in many clinical scenarios, which limits the applicability of traditional unsupervised anomaly detection methods. D²FAD employs a dual distillation strategy: 1) Teacher-Student Distillation, where a student decoder learns to reconstruct multi-scale features from a pre-trained (and frozen) teacher encoder on query images, and 2) Student Self-Distillation, where the student decoder learns to produce consistent representations between a query image and a small support set of normal images from the same domain. Additionally, the authors propose a "learn-to-weight" mechanism to dynamically assign importance to different support images based on their relevance to the query. The authors also contribute a new comprehensive benchmark dataset for medical FAD, curated from several public sources. Experiments show that D²FAD significantly outperforms a wide range of unsupervised and few-shot anomaly detection methods across this new benchmark.

**Strengths:**

Clear Motivation and Problem Formulation: The paper does an excellent job of motivating the need for few-shot anomaly detection in clinical practice, grounding the research in a real-world problem. The formalization of the FAD task is clear and precise.

Elegant and Effective Method: The D²FAD framework is simple yet powerful. The dual distillation concept is intuitive and well-justified. By using a frozen pre-trained encoder as the teacher, the method is parameter-efficient and avoids the need for complex architectural designs.

Thorough and Convincing Experiments: The empirical evaluation is a major strength. The authors have not only proposed a new method but have also built a strong benchmark to validate it. The comparisons against a wide array of baselines are comprehensive, and the ablation studies provide strong evidence for the efficacy of each component of their proposed method. The consistently high performance across multiple datasets and few-shot settings (2, 4, 8-shot) is very impressive.

High Practical Relevance: The method is well-aligned with clinical reality, where often only a few trusted normal cases are available for reference. The high AUROC scores, combined with the method's efficiency (as shown in Figure 3), suggest strong potential for practical deployment.

**Weaknesses:**

Limited Technical Depth in "Learn-to-Weight": While the "learn-to-weight" mechanism (Eq. 4) is a good idea, its presentation is somewhat brief. It is essentially a scaled dot-product attention between the query and support features. The paper could benefit from a deeper analysis or discussion of this component. For example, are there other ways to instantiate this weighting? How does this mechanism behave in practice (e.g., does it learn to ignore outlier-like support images)?

Sensitivity to the Pre-trained Teacher: The entire framework's performance is heavily dependent on the quality of the frozen teacher encoder (pre-trained on ImageNet). While the ablation in Table 4 explores different backbones, it raises a question: how does the choice of pre-training dataset (not just architecture) affect performance? Would a teacher pre-trained on a large medical dataset (if available) perform even better? A brief discussion on the limitations imposed by the teacher's domain (natural images vs. medical) would strengthen the paper.

"Simplicity" as a Double-Edged Sword: While the architectural simplicity is a strength, some might argue that the individual components (knowledge distillation, self-distillation, attention) are themselves not new. The novelty lies in their specific combination for the FAD task. This is a minor point, as the combination is non-trivial and highly effective, but it is worth noting that the paper's contribution is more about a novel framework than a fundamentally new algorithm.

**Questions:**

Regarding the "Learn-to-Weight" Mechanism: This is an interesting component. Could you provide some qualitative analysis to help understand what it learns? For instance, if you provide a "bad" or less relevant normal image in the support set, does the model learn to assign it a lower weight w_i? Visualizing these weights for a few examples could provide valuable insight.

Impact of Teacher Model: Your results in Table 4 show that WideResNet-50 is the best teacher. This is consistent with many distillation-based anomaly detection papers. However, you also note that very large models like Swin Transformers perform suboptimally. Do you have an intuition for why this "representational gap" occurs? Is it because the student decoder is too simple to distill from such a complex teacher, or is there another reason?


Dataset Details: Thank you for curating this excellent benchmark. Could you clarify if the train/test splits for the leave-one-out protocol are strictly separated by dataset source (e.g., train on HIS, LAG, APTOS, RSNA; test on Brain Tumor) or by anatomy/modality? For instance, if you train on all datasets except Brain Tumor (MRI), would you also exclude other MRI datasets from the training set to test for modality generalization?

---

> ### Author Response · Authors · 2025-11-27
>
> >Limited Technical Depth in “Learn-to-Weight”: While the “learn-to-weight” mechanism (Eq. 4) is a good idea, its presentation is somewhat brief. It is essentially a scaled dot-product attention between the query and support features. The paper could benefit from a deeper analysis or discussion of this component. For example, are there other ways to instantiate this weighting? How does this mechanism behave in practice (e.g., does it learn to ignore outlier-like support images)?
>
> Many thanks for the thoughtful comments and valuable suggestions. As recommended, we have conducted a more in-depth exploration in the revised version (see Appendix F), focusing on two aspects.
>
> 1. We examined several additional ways to compute the weights for reference samples and added the corresponding experiments in the revision.
>
> 2. We included examples illustrating how the mechanism distributes weights across support images. The visualizations clearly show that the model automatically assigns lower weights to support samples that are less relevant to the query.
> ---
> “To assess the flexibility of our weighting mechanism, we introduce three alternative similarity functions—Gaussian, embedded Gaussian, and concatenation—to compute query–support similarity, followed by softmax normalization:
>
> ### Gaussian
>
> $$
> w_{i}=\mathrm{softmax}(e^{z^i_{\mathtt{qry}}(z^i_{\mathtt{spt}})^\mathrm{T}})
> $$
>
> Euclidean distance, as in [1], is also applicable, but the dot product is more implementation-friendly in modern deep-learning frameworks.
>
> ### Embedded Gaussian
>
> $$
> w_{i}=\mathrm{softmax}(e^{\theta(z^i_{\mathtt{qry}}){\phi(z^i_{\mathtt{spt}})}^\mathrm{T}})
> $$
>
> where $\theta$ and $\phi$ are linear projections implemented with $1\times1$ convolutions.
>
> ### Concatenation
>
> $$
> w_{i}=\mathrm{softmax}(\mathrm{ReLU}(w^\mathrm{T}[{\theta(z^i_{\mathtt{qry}}),{\phi(z^i_{\mathtt{spt}})}}]))
> $$
>
> where $[\cdot,\cdot]$ denotes concatenation, and $w$ is a learnable weight vector. The query feature $z^i_{\mathtt{qry}} \in \mathbb{R}^{1\times C_i\times H_i\times W_i}$ is broadcast to $\mathbb{R}^{K\times (C_i\times H_i\times W_i)}$ for this computation.
>
> The remaining steps follow Section 2.3. As shown in Table 7, different weighting strategies produce comparable but distinct results. The scaled dot-product formulation used in our main model shows slightly better performance, likely because it provides a more direct and effective measure of feature similarity. These variants demonstrate the flexibility of our learn-to-weight design, and other alternatives may further improve performance.
>
> We also visualize the learned weights in Figure 6. Support images that are normal, anomalous, or semantically irrelevant are marked in green, red, and yellow, respectively. Across multiple layers (weight 1/2/3), the model consistently assigns lower weights to anomalous or irrelevant support images, highlighting the effectiveness of the proposed mechanism.”
>
> [1]Buades, Antoni, Coll, Bartomeu, and Morel, J.-M. A non-local algorithm for image denoising. In Proceedings of the IEEE/CVF Conference on Computer Vision and Pattern Recognition (CVPR), volume 2, pages 60–65, 2005.
>
> Tabel 7: Performance comparison among different instantiations of the learn-to-weight mechanism.
> ||Gaussian|Gaussian, embed|Concatenation|Ours|
> |:----|:------:|:-------------:|:-----------:|:--:|
> |**HIS 2-shot**|95.7|96.0|95.9|**96.7**|
> |**HIS 4-shot**|97.4|**98.6**|97.6|98.4|
> |**HIS 8-shot**|98.0|98.4|98.2|**99.0**|
> |**LAG 2-shot**|95.6|96.9|96.7|**97.6**|
> |**LAG 4-shot**|96.9|97.2|97.1|**97.9**|
> |**LAG 8-shot**|97.0|97.4|97.7|**98.0**|
> |**APTOS 2-shot**|99.9|99.9|99.9|**100.0**|
> |**APTOS 4-shot**|99.2|**100.0**|99.8|**100.0**|
> |**APTOS 8-shot**|**100.0**|**100.0**|**100.0**|**100.0**|
> |**RSNA 2-shot**|97.0|98.8|97.5|**99.0**|
> |**RSNA 4-shot**|98.1|98.7|97.9|**99.1**|
> |**RSNA 8-shot**|98.8|99.2|99.0|**99.8**|
> |**Brain Tumor 2-shot**|94.1|94.5|94.3|**95.1**|
> |**Brain Tumor 4-shot**|95.3|95.7|94.5|**96.3**|
> |**Brain Tumor 8-shot**|95.8|96.2|96.0|**96.8**|

---

> ### Author Response · Authors · 2025-11-27
>
> >Sensitivity to the Pre-trained Teacher: The entire framework’s performance is heavily dependent on the quality of the frozen teacher encoder (pre-trained on ImageNet). While the ablation in Table 4 explores different backbones, it raises a question: how does the choice of pre-training dataset (not just architecture) affect performance? Would a teacher pre-trained on a large medical dataset (if available) perform even better? A brief discussion on the limitations imposed by the teacher’s domain (natural images vs. medical) would strengthen the paper.
>
> We thank the reviewer for the insightful comments. To address this concern, we conducted an additional experiment using a ResNet-50 teacher pre-trained on a large-scale medical dataset (RadImageNet), and compared it with the same architecture pre-trained on ImageNet. The medical-domain teacher yields consistent improvements across all five datasets, with an average AUROC gain of 0.91%. This result highlights the benefit of aligning the teacher’s pre-training domain with the target task. We have added these results and the accompanying discussion in Appendix B of the revised version.
>
> “*To examine how the teacher’s pre-training domain influences performance, we compare ResNet-50 models trained on ImageNet (natural images) and RadImageNet (medical images). As shown in Table 6, the medically pre-trained teacher provides an average improvement of 0.91%, indicating that domain-aligned representations enhance accuracy. This also reinforces our main claim that the proposed distillation framework is flexible and can readily incorporate stronger or domain-specific teacher models.*”
>
> Table 6: Performance comparison between different pre-training domains using the ResNet-50 backbone.
> ||ImageNet|RadImageNet|
> |:------------------|:------:|:---------:|
> |**HIS 2-shot**|91.4|**93.6**|
> |**HIS 4-shot**|93.9|**95.2**|
> |**HIS 8-shot**|95.4|**95.6**|
> |**LAG 2-shot**|92.5|**93.0**|
> |**LAG 4-shot**|**96.3**|95.9|
> |**LAG 8-shot**|96.4|**96.8**|
> |**APTOS 2-shot**|**100.0**|99.9|
> |**APTOS 4-shot**|**100.0**|**100.0**|
> |**APTOS 8-shot**|**100.0**|**100.0**|
> |**RSNA 2-shot**|94.2|**97.4**|
> |**RSNA 4-shot**|97.1|**98.0**|
> |**RSNA 8-shot**|98.3|**98.6**|
> |**Brain Tumor 2-shot**|94.6|**95.3**|
> |**Brain Tumor 4-shot**|95.3|**97.7**|
> |**Brain Tumor 8-shot**|96.4|**98.4**|

---

> ### Author Response · Authors · 2025-11-27
>
> >“Simplicity” as a Double-Edged Sword: While the architectural simplicity is a strength, some might argue that the individual components (knowledge distillation, self-distillation, attention) are themselves not new. The novelty lies in their specific combination for the FAD task. This is a minor point, as the combination is non-trivial and highly effective, but it is worth noting that the paper’s contribution is more about a novel framework than a fundamentally new algorithm.
>
> Many thanks for the thoughtful comment. We agree with the reviewer’s point. As suggested, we now clarify in the revised version that our contribution lies in proposing a simple yet effective framework tailored for FAD, rather than introducing a fundamentally new algorithm.

---

> ### Author Response · Authors · 2025-11-27
>
> >Impact of Teacher Model: Your results in Table 4 show that WideResNet-50 is the best teacher. This is consistent with many distillation-based anomaly detection papers. However, you also note that very large models like Swin Transformers perform suboptimally. Do you have an intuition for why this “representational gap” occurs? Is it because the student decoder is too simple to distill from such a complex teacher, or is there another reason?
>
> We greatly thank the reviewer for the thoughtful question. Below is our analysis.
>
> 1. Since the student has a mirrored architecture of the teacher, it is unlikely that the underperformance is caused by the student decoder being “too simple”.
>
> 2. We believe the gap primarily arises from the nature of Transformer representations. Compared with CNNs, Transformer features tend to be more abstract and highly semantic. Prior studies have noted that such representations can be harder to transfer through distillation, because reproducing the teacher's complex attention patterns and global dependencies can be challenging. In contrast, CNN teachers provide more structured and easier-to-distill feature hierarchies. As a result, very large Transformer teachers such as Swin may produce richer but less student-friendly representations, leading to the observed performance gap.

---

> ### Author Response · Authors · 2025-11-27
>
> >Dataset Details: Thank you for curating this excellent benchmark. Could you clarify if the train/test splits for the leave-one-out protocol are strictly separated by dataset source (e.g., train on HIS, LAG, APTOS, RSNA; test on Brain Tumor) or by anatomy/modality? For instance, if you train on all datasets except Brain Tumor (MRI), would you also exclude other MRI datasets from the training set to test for modality generalization?
>
> Sorry about not being clear on this point. In our benchmark, the “leave-one-out” protocol is strictly defined by dataset source (or equivalently, by task). Each dataset (HIS, LAG, APTOS, RSNA, Brain Tumor) corresponds to a different task. For example, when the Brain Tumor dataset is held out for testing, the training set consists of all remaining datasets (HIS, LAG, APTOS, RSNA). We do not exclude datasets based on imaging modality, because datasets with the same modality may still involve different tasks.

---

### Official Review · Reviewer_fFwa · 2025-10-31

**Soundness:** 3
**Presentation:** 3
**Contribution:** 2
**Rating:** 6
**Confidence:** 3

**Summary:**

This paper proposes $D^24FAD$, a dual-distillation framework for few-shot anomaly detection in medical imaging. The method employs a pre-trained encoder on ImageNet as a teacher and a learnable decoder as a student. The student distills knowledge from the teacher on query images and performs self-distillation on a few normal support images, creating a compact representation of “normal” anatomy for each task. A learn-to-weight module dynamically adjusts the contribution of each support sample conditioned on the query. The approach is trained only on normal data and evaluated in a few-shot setting (2-, 4-, and 8-shot) across five medical datasets spanning different modalities and organs. Experiments show consistently good AUROC results and competitive inference efficiency.

**Strengths:**

1. The paper introduces a clear and well-motivated dual-distillation framework ($D^24FAD$) for few-shot anomaly detection, combining a teacher–student distillation mechanism with an additional student self-distillation path and a learn-to-weight module that adaptively re-weights support images conditioned on the query. While the core components (knowledge distillation, few-shot learning) are known, their integration into a unified few-shot medical anomaly detection framework is novel and conceptually elegant. The problem formulation, detecting unseen anomalies from only a few normal references, addresses a challenging and underexplored setting in medical imaging.

2. The work is technically solid and empirically well supported. The authors evaluate their approach on five heterogeneous medical datasets and include ablation studies that clearly demonstrate the contribution of each component. The training procedure is reproducible and relies on standard architectures and optimization methods. The model achieves consistently strong performance, often surpassing baselines by a substantial margin, while maintaining efficient inference and low memory consumption. It should be noted, however, that several competing methods (e.g., PatchCore, RD4AD) were originally designed for pixel-level anomaly localization. In this paper they are evaluated only at the image level, which might underestimate their actual capability. Consequently, while the reported results indicate strong performance, the comparative advantage should be interpreted with some caution.

3. The paper is clearly organized and readable. The motivation, methodology, and evaluation are logically connected.

4. The paper targets an important real-world problem, automatic anomaly detection under limited supervision in medical imaging, which has substantial clinical implications.

**Weaknesses:**

The main limitation of the paper lies in the formulation of the task. Although the work is presented as addressing few-shot anomaly detection, the evaluation is restricted to image-level AUROC, effectively turning the problem into a binary classification task (normal versus abnormal). While the model internally produces anomaly maps, no quantitative localization results are provided (e.g., Dice, IoU, or AUPRO). This simplification reduces the methodological complexity of the problem and limits the demonstrated clinical applicability of the approach, where spatial localization of pathological regions is often essential.

A second concern relates to the evaluation of competing methods. Several baselines included in the comparison, such as PatchCore and RD4AD, were originally designed for pixel-level anomaly localization. Converting these approaches to image-level evaluation may underestimate their true capability and potentially exaggerate the reported relative gains of the proposed model. A clearer justification or separate localization-based comparison would strengthen the empirical claims.

In terms of experimental details, several important aspects remain unspecified. The procedure used to select the weighting coefficient λ (Table 3) is not described; it is unclear whether this was tuned using a validation set or chosen post-hoc on test performance, which risks overfitting on test data. Similarly, the results reported in Figure 3 for inference time and memory usage are not linked to a particular dataset, making the comparison less transparent.

The framework’s architectural flexibility also appears limited. Because the distillation losses are computed layer-wise between teacher and student features of matching dimensions, both networks must share a compatible structure. The paper does not mention whether any projection or adaptation modules were used to support heterogeneous architectures.

Finally, the analysis remains confined to a teacher pretrained on ImageNet, which introduces a potential domain mismatch for medical images. The effect of employing a medical-domain pretrained backbone (e.g. BioMedCLIP) is not investigated. Such an analysis would provide valuable insight into the robustness and domain generalization of the proposed framework.

**Questions:**

1.Task formulation and localization:
Can you provide quantitative results for anomaly localization (e.g., Dice, IoU) to demonstrate the spatial detection ability of your method? If such annotations are not available, could you at least show quantitative proxy metrics or additional qualitative evidence supporting the localization quality shown in Appendix B?

2. Evaluation of segmentation-based baselines:
Several baselines such as PatchCore and RD4AD were originally designed for pixel-level anomaly localization. How were these methods adapted to the image-level AUROC evaluation used in this paper? Could you clarify whether their localization outputs were averaged or thresholded, and how this might affect the reported results?

3. Weight coefficient λ selection:
How was the optimal value of λ = 0.1 (Table 3) determined? Was there a validation set or a held-out meta-task used for tuning, or was the selection based on post-hoc test performance? A clear description of this procedure would help assess the fairness of comparisons.

4. Inference time and memory usage (Figure 3):
For which dataset(s) were the inference time and memory measurements obtained? Are the reported values averaged across datasets or computed on a single representative dataset?

5. Architectural flexibility:
Does the proposed framework require the teacher and student networks to share an identical architecture and layer resolution? If not, are there any projection or adaptation modules that enable distillation between heterogeneous backbones?

6. Pretraining domain and robustness:
Have you considered evaluating the method with a medical-domain pretrained teacher network (e.g.,  BioMedCLIP)? Such experiments would clarify whether the reliance on ImageNet pretraining affects the generalization ability across medical modalities.

---

> ### Author Response · Authors · 2025-11-27
>
> >The main limitation of the paper lies in the formulation of the task. Although the work is presented as addressing few-shot anomaly detection, the evaluation is restricted to image-level AUROC, effectively turning the problem into a binary classification task (normal versus abnormal). While the model internally produces anomaly maps, no quantitative localization results are provided (e.g., Dice, IoU, or AUPRO). This simplification reduces the methodological complexity of the problem and limits the demonstrated clinical applicability of the approach, where spatial localization of pathological regions is often essential.
>
> Thanks a lot for the comment. The primary focus of this work is image-level anomaly detection. To make few-shot medical anomaly detection meaningful, we intentionally collect datasets that cover diverse organs and imaging modalities. However, this diversity also makes it difficult to ensure that all datasets come with pixel-wise segmentation masks. Moreover, obtaining accurate pixel-level annotations is challenging for many medical imaging datasets [1]. For example, in breast cancer histopathology (HIS), the distinction between malignant and normal regions often depends on staining intensity and cellular density [2], which can manifest globally or in heterogeneous patterns, making precise pixel-wise labeling impractical. In such scenarios, image-level detection itself provides clinically useful signals, especially for screening and triage.
>
> This setting is also consistent with common practice in the few-shot anomaly detection literature, where many works either do not report pixel-level metrics or report them only on subsets of data with available annotations [3,4,5,6]. Following these established practices, we adopt image-level AUROC as our primary evaluation metric.
>
> We have now made this limitation explicit in the revised version and noted that incorporating pixel-level localization metrics would be an important direction for future work (see Appendix E).
>
> “*Despite promising qualitative localization, the absence of pixel-level annotations in our benchmark hinders quantitative evaluation, and this limitation could be addressed with fully annotated datasets in the future.*”
>
> [1]Yu Cai, Weiwen Zhang, Hao Chen, and Kwang-Ting Cheng. MedIAnomaly: A comparative study of anomaly detection in medical images. Medical Image Analysis, 102:103500, 2025.
>
> [2]Veta, M., Pluim, J. P., Van Diest, P. J., & Viergever, M. A. Breast cancer histopathology image analysis: A review. IEEE Transactions on Biomedical Engineering, 61(5), 1400–1411, 2014.
>
> [3]Chaoqin Huang, Haoyan Guan, Aofan Jiang, Ya Zhang, Michael Spratling, and Yan-Feng Wang. Registration based few-shot anomaly detection. In European Conference on Computer Vision (ECCV), pages 303–319, 2022.
>
> [4]Jiawen Zhu and Guansong Pang. Toward generalist anomaly detection via in-context residual learning with few-shot sample prompts. In Proceedings of the IEEE/CVF Conference on Computer Vision and Pattern Recognition (CVPR), pages 17826–17836, 2024.
>
> [5]Guansong Pang, Chunhua Shen, Huidong Jin, and Anton Van Den Hengel. Deep weakly-supervised anomaly detection. In Proceedings of the 29th ACM SIGKDD Conference on Knowledge Discovery and Data Mining (KDD), pages 1795–1807, 2023.
>
> [6]Ximiao Zhang, Min Xu, Dehui Qiu, Ruixin Yan, Ning Lang, and Xiuzhuang Zhou. MediCLIP: Adapting CLIP for few-shot medical image anomaly detection. In International Conference on Medical Image Computing and Computer-Assisted Intervention (MICCAI), pages 458–468, 2024.

---

> ### Author Response · Authors · 2025-11-27
>
> >Evaluation of segmentation-based baselines: Several baselines such as PatchCore and RD4AD were originally designed for pixel-level anomaly localization. How were these methods adapted to the image-level AUROC evaluation used in this paper? Could you clarify whether their localization outputs were averaged or thresholded, and how this might affect the reported results?
>
> We appreciate the reviewer’s questions.
>
> For both PatchCore and RD4AD, we strictly follow the image-level AUROC evaluation defined in their original papers and official code. These methods derive an image-level anomaly score by max-pooling their pixel-level anomaly maps, and we directly adopt this procedure without any modification. As far as we know, this aggregation strategy is a widely accepted practice in the anomaly detection community and does not introduce any known negative effect.
>
> To clarify this in the paper, we have added the following description in the revised manuscript:
>
> “*Note that some baselines, such as PatchCore and RD4AD, compute image-level anomaly scores by applying max-pooling over the pixel-level anomaly maps.*”

---

> ### Author Response · Authors · 2025-11-27
>
> >Weight coefficient λ selection: How was the optimal value of λ = 0.1 (Table 3) determined? Was there a validation set or a held-out meta-task used for tuning, or was the selection based on post-hoc test performance? A clear description of this procedure would help assess the fairness of comparisons.
>
> Sorry about not being clear enough on these points. The optimal value of $\lambda$ = 0.1 was selected through empirical tuning on a validation set. We observed the same trend on the test set, as reported in Table 3. We have clarified this in the revised version (see Section 4.3).
>
> “*The weight coefficient $\lambda$ balances the contribution between teacher-student distillation and student self-distillation in the overall objective function. To determine its optimal value, we conducted a grid search over $\lambda \in $ {1.0, 0.5, 0.1, 0.05, 0.01} using a dedicated validation set containing 100 normal and 100 anomalous images per task. Results in Table 3 indicate that setting $\lambda$ = 0.1 yields the best average performance across all datasets on the test set.*”

---

> ### Author Response · Authors · 2025-11-27
>
> >Inference time and memory usage (Figure 3): For which dataset(s) were the inference time and memory measurements obtained? Are the reported values averaged across datasets or computed on a single representative dataset?
>
> Thanks a lot for the questions. The measurements were obtained across all five datasets (HIS, LAG, APTOS, RSNA, and Brain Tumor) and the reported values represent averages. We have clarified this in the revised version (see Section 4.5).
>
> “*We compare our model with competing few-shot anomaly detection methods and a representative unsupervised method (RD4AD) in terms of AUROC, inference time, and memory usage during inference, averaged across all five datasets.*”

---

> ### Author Response · Authors · 2025-11-27
>
> >Architectural flexibility: Does the proposed framework require the teacher and student networks to share an identical architecture and layer resolution? If not, are there any projection or adaptation modules that enable distillation between heterogeneous backbones?
>
> We appreciate the reviewer for raising this interesting point. With full respect, we actually believe our framework is already quite flexible. In practice, given any pre-trained network as the teacher, we simply use its mirror as the student. This straightforward setup requires no modifications to the network and works well. At the moment, we do not see a need for a heterogeneous student network, which would introduce extra complexity. Moreover, almost all distillation-based methods for anomaly detection adopt a homogeneous design [1,2,3,4]. In principle, a heterogeneous design could be feasible, since distillation operates at the feature level, as long as the overall architecture—e.g., convolutional blocks or Transformer layers—is compatible.
>
> [1]Mohammadreza Salehi, Niousha Sadjadi, Soroosh Baselizadeh, Mohammad H. Rohban, and Hamid R. Rabiee. Multiresolution knowledge distillation for anomaly detection. In Proceedings of the IEEE/CVF Conference on Computer Vision and Pattern Recognition (CVPR), pages 14902-14912, 2021.
>
> [2]Paul Bergmann, Michael Fauser, David Sattlegger, and Carsten Steger. Uninformed students: Student-teacher anomaly detection with discriminative latent embeddings. In Proceedings of the IEEE/CVF Conference on Computer Vision and Pattern Recognition (CVPR), pages 4183-4192, 2020.
>
> [3]Guodong Wang, Shumin Han, Errui Ding, and Di Huang. Student-teacher feature pyramid matching for anomaly detection. In Proceedings of the IEEE/CVF Conference on Computer Vision and Pattern Recognition (CVPR), pages 14902-14912, 2021.
>
> [4]Zhihao Gu, Liang Liu, Xu Chen, Ran Yi, Jiangning Zhang, Yabiao Wang, Chengjie Wang, Annan Shu, Guannan Jiang, and Lizhuang Ma. Remembering normality: Memory-guided knowledge distillation for unsupervised anomaly detection. In Proceedings of the IEEE/CVF International Conference on Computer Vision (ICCV), pages 16401-16409, 2023.

---

> ### Author Response · Authors · 2025-11-27
>
> >Pretraining domain and robustness: Have you considered evaluating the method with a medical-domain pretrained teacher network (e.g., BioMedCLIP)? Such experiments would clarify whether the reliance on ImageNet pretraining affects the generalization ability across medical modalities.
>
> Thank a lot for the question. Following the reviewer’s advice, we have added experiments using a medical-domain teacher network (BioMedCLIP) as well as another pre-trained network, CLIP-ViT. The results are now reported in Table 4 of the revised version.
>
> Table 4: Quantitative performance comparison of various backbone architectures employed as the teacher network across all datasets.
> ||CLIP|BiomedCLIP|
> |:------------------|:----:|:----------:|
> |**HIS 2-shot**|88.5|**89.2**|
> |**HIS 4-shot**|91.2|**92.1**|
> |**HIS 8-shot**|93.5|**94.3**|
> |**LAG 2-shot**|90.3|**91.5**|
> |**LAG 4-shot**|93.1|**95.2**|
> |**LAG 8-shot**|93.8|**95.7**|
> |**APTOS 2-shot**|97.8|**99.9**|
> |**APTOS 4-shot**|**99.9**|**99.9**|
> |**APTOS 8-shot**|99.9|**100.0**|
> |**RSNA 2-shot**|85.7|**91.2**|
> |**RSNA 4-shot**|90.4|**94.3**|
> |**RSNA 8-shot**|93.1|**96.2**|
> |**Brain Tumor 2-shot**|**92.8**|91.7|
> |**Brain Tumor 4-shot**|93.5|**94.6**|
> |**Brain Tumor 8-shot**|95.2|**98.1**|

---

### Official Review · Reviewer_gRTX · 2025-11-02

**Soundness:** 3
**Presentation:** 3
**Contribution:** 3
**Rating:** 4
**Confidence:** 5

**Summary:**

The paper proposes a dual-distillation framework for few-shot anomaly detection in medical images. The method freezes an ImageNet-pretrained encoder as a “teacher” and trains a lightweight decoder as a “student,” combining (1) feature distillation on query images and (2) self-distillation on support images. A 5-dataset benchmark (13,084 images) is introduced, and the method is shown to outperform prior work in image-level AUROC under 2/4/8-shot settings.

**Strengths:**

- Clear motivation and task definition for few-shot anomaly detection in medical settings.
- Architecture is simple, fast, and avoids large generative models.
- Strong image-level AUROC across multiple datasets and shot settings.

**Weaknesses:**

- The work repeatedly emphasizes “dual distillation” as a key contribution, but the process does not fully match established definitions of distillation in the literature. Since the teacher network is frozen, and the student is not learning logits or semantic knowledge but merely reconstructing features, the term distillation may be overstated. This weakens the conceptual positioning of the contribution: the method is an anomaly-detection reconstruction framework rather than a genuine knowledge-transfer framework. A more precise formulation would strengthen the paper’s technical clarity.
- The fairness of the baseline comparison is still unclear. Although it is likely that MediCLIP, MVFA, and INP-Former are evaluated using their official pretrained checkpoints followed by few-shot inference (rather than re-trained from scratch), this introduces a different concern: the pretraining data used for these models may not be aligned with the data available to the proposed method. Since these baselines were originally trained on large external datasets (e.g., CLIP pretraining or full normal medical datasets), it is important to clarify whether (1) their pretraining data overlaps with the test datasets, and (2) whether the proposed method has access to comparable pretraining resources. Without such clarification, the performance difference may reflect differences in dataset exposure rather than architectural effectiveness.
- Only image-level detection is evaluated, no localization metrics. Pixel-level anomaly localization (e.g., heatmaps, PRO, Dice, pixel-AUROC) is essential for clinical use. Several baselines do support localization (e.g., MVFA, AnomalyGPT), but the paper does not report or discuss localization performance, making the clinical impact claim incomplete.
- Inconsistent reporting of “Ours” across tables. The AUROC results for the proposed method in Table 1 are lower than those in the final row of Table 2, even though both appear to evaluate the same benchmark. This suggests the two tables use different configurations (e.g., stronger backbone, more loss terms, or ablation-optimized settings), but this is not stated anywhere. It becomes unclear which version represents the “main result,” and whether the strongest performance was omitted from the primary comparison.

**Questions:**

- Clarify the evaluation setting for MediCLIP, MVFA, and INP-Former
- Does the method support pixel-level localization? If yes, why are localization metrics not reported?
- Why does Table 1 report lower performance than the last row of Table 2?
- Have you experimented with stronger pretrained teachers (e.g., CLIP-ViT) instead of ImageNet CNNs?

---

> ### Author Response · Authors · 2025-11-27
>
> >The work repeatedly emphasizes “dual distillation” as a key contribution, but the process does not fully match established definitions of distillation in the literature. Since the teacher network is frozen, and the student is not learning logits or semantic knowledge but merely reconstructing features, the term distillation may be overstated. This weakens the conceptual positioning of the contribution: the method is an anomaly-detection reconstruction framework rather than a genuine knowledge-transfer framework. A more precise formulation would strengthen the paper’s technical clarity.
>
> Many thanks for the comment.
>
> Our usage of the term “distillation” follows the convention in several prior anomaly detection works [1,2,3,4,5,6], which also adopt a fixed pre-trained teacher and train a student network to align or reconstruct features. Within this community, such approaches are commonly referred to as distillation-based methods. With full respect to the reviewer’s viewpoint, we believe the term remains appropriate in the context of anomaly detection.
>
> Moreover, distillation in the broader machine learning literature is not restricted to transferring logits or semantic information. Feature-level distillation is a well-recognized form of knowledge transfer [7], and our formulation aligns with this perspective.
>
> To maintain continuity with established terminology in the field, we keep the term “distillation”. At the same time, in response to the reviewer’s valuable comment, we have revised the paper to explicitly clarify this point to avoid potential confusion for both the reviewer and future readers.
>
> “*Note that some works refer to this process as feature reconstruction. In this work, for consistency with prior anomaly detection methods [1,2,3,4,5,6], we use the term distillation.*”
>
> [1]Mohammadreza Salehi, Niousha Sadjadi, Soroosh Baselizadeh, Mohammad H. Rohban, and Hamid R. Rabiee. Multiresolution knowledge distillation for anomaly detection. In Proceedings of the IEEE/CVF Conference on Computer Vision and Pattern Recognition (CVPR), pages 14902–14912, 2021.
>
> [2]Hanqiu Deng and Xing Li. Anomaly detection via reverse distillation from one-class embedding. In Proceedings of the IEEE/CVF Conference on Computer Vision and Pattern Recognition (CVPR), pages 9737–9746, 2022.
>
> [3]Liyi Yao and Shaobing Gao. Dual-student knowledge distillation networks for unsupervised anomaly detection. In Proceedings of the IEEE/CVF Conference on Computer Vision and Pattern Recognition (CVPR), pages 14902–14912, 2024.
>
> [4]Zhihao Gu, Liang Liu, Xu Chen, Ran Yi, Jiangning Zhang, Yabiao Wang, Chengjie Wang, Annan Shu, Guannan Jiang, and Lizhuang Ma. Remembering normality: Memory-guided knowledge distillation for unsupervised anomaly detection. In Proceedings of the IEEE/CVF International Conference on Computer Vision (ICCV), pages 16401–16409, 2023.
>
> [5]Ye Ma, Xu Jiang, Nan Guan, and Wang Yi. Anomaly detection based on multi-teacher knowledge distillation. Journal of Systems Architecture, 138:102861, 2023.
>
> [6]Qihang Zhou, Shibo He, Haoyu Liu, Tao Chen, and Jiming Chen. Pull & push: Leveraging differential knowledge distillation for efficient unsupervised anomaly detection and localization. IEEE Transactions on Circuits and Systems for Video Technology, 33(5):2176–2189, 2022.
>
> [7]Adriana Romero, Nicolas Ballas, Samira Ebrahimi Kahou, Antoine Chassang, Carlo Gatta, and Yoshua Bengio. FitNets: Hints for thin deep nets. In International Conference on Learning Representations (ICLR), 2015.

---

> ### Author Response · Authors · 2025-11-27
>
> >The fairness of the baseline comparison is still unclear. Although it is likely that MediCLIP, MVFA, and INP-Former are evaluated using their official pretrained checkpoints followed by few-shot inference (rather than re-trained from scratch), this introduces a different concern: the pretraining data used for these models may not be aligned with the data available to the proposed method. Since these baselines were originally trained on large external datasets (e.g., CLIP pretraining or full normal medical datasets), it is important to clarify whether (1) their pretraining data overlaps with the test datasets, and (2) whether the proposed method has access to comparable pretraining resources. Without such clarification, the performance difference may reflect differences in dataset exposure rather than architectural effectiveness.
>
> Thanks for raising this important concern, and we apologize for not making this clear enough in the paper. For a fair comparison, all models—including MediCLIP, MVFA, and INP-Former—are trained from scratch on the same data split. That is, we do not use their official pre-trained checkpoints for few-shot inference. Instead, we follow their official implementations and re-train each model. We have clarified this more explicitly in the revised version (see Section 4.1).
>
> “*For unsupervised methods and few-shot methods without generalization capability (MediCLIP, MVFA, and INP-Former), we utilize K support (normal) images as training data to train the models and evaluate them on the same task.*”

---

> ### Author Response · Authors · 2025-11-27
>
> >Only image-level detection is evaluated, no localization metrics. Pixel-level anomaly localization (e.g., heatmaps, PRO, Dice, pixel-AUROC) is essential for clinical use. Several baselines do support localization (e.g., MVFA, AnomalyGPT), but the paper does not report or discuss localization performance, making the clinical impact claim incomplete.
>
> Thanks a lot for the comment. The primary focus of this work is image-level anomaly detection. To make few-shot medical anomaly detection meaningful, we intentionally collect datasets that cover diverse organs and imaging modalities. However, this diversity also makes it difficult to ensure that all datasets come with pixel-wise segmentation masks. Moreover, obtaining accurate pixel-level annotations is challenging for many medical imaging datasets [1]. For example, in breast cancer histopathology (HIS), the distinction between malignant and normal regions often depends on staining intensity and cellular density [2], which can manifest globally or in heterogeneous patterns, making precise pixel-wise labeling impractical. In such scenarios, image-level detection itself provides clinically useful signals, especially for screening and triage.
>
> This setting is also consistent with common practice in the few-shot anomaly detection literature, where many works either do not report pixel-level metrics or report them only on subsets of data with available annotations [3,4,5,6]. Following these established practices, we adopt image-level AUROC as our primary evaluation metric.
>
> We have now made this limitation explicit in the revised version and noted that incorporating pixel-level localization metrics would be an important direction for future work (see Appendix E).
>
> “*Despite promising qualitative localization, the absence of pixel-level annotations in our benchmark hinders quantitative evaluation, and this limitation could be addressed with fully annotated datasets in the future.*”
>
> [1]Yu Cai, Weiwen Zhang, Hao Chen, and Kwang-Ting Cheng. MedIAnomaly: A comparative study of anomaly detection in medical images. Medical Image Analysis, 102:103500, 2025.
>
> [2]Veta, M., Pluim, J. P., Van Diest, P. J., & Viergever, M. A. Breast cancer histopathology image analysis: A review. IEEE Transactions on Biomedical Engineering, 61(5), 1400–1411, 2014.
>
> [3]Chaoqin Huang, Haoyan Guan, Aofan Jiang, Ya Zhang, Michael Spratling, and Yan-Feng Wang. Registration based few-shot anomaly detection. In European Conference on Computer Vision (ECCV), pages 303–319, 2022.
>
> [4]Jiawen Zhu and Guansong Pang. Toward generalist anomaly detection via in-context residual learning with few-shot sample prompts. In Proceedings of the IEEE/CVF Conference on Computer Vision and Pattern Recognition (CVPR), pages 17826–17836, 2024.
>
> [5]Guansong Pang, Chunhua Shen, Huidong Jin, and Anton Van Den Hengel. Deep weakly-supervised anomaly detection. In Proceedings of the 29th ACM SIGKDD Conference on Knowledge Discovery and Data Mining (KDD), pages 1795–1807, 2023.
>
> [6]Ximiao Zhang, Min Xu, Dehui Qiu, Ruixin Yan, Ning Lang, and Xiuzhuang Zhou. MediCLIP: Adapting CLIP for few-shot medical image anomaly detection. In International Conference on Medical Image Computing and Computer-Assisted Intervention (MICCAI), pages 458–468, 2024.

---

> ### Author Response · Authors · 2025-11-27
>
> >Inconsistent reporting of “Ours” across tables. The AUROC results for the proposed method in Table 1 are lower than those in the final row of Table 2, even though both appear to evaluate the same benchmark. This suggests the two tables use different configurations (e.g., stronger backbone, more loss terms, or ablation-optimized settings), but this is not stated anywhere. It becomes unclear which version represents the “main result,” and whether the strongest performance was omitted from the primary comparison.
>
> We thank the reviewer for pointing out this issue. The results in Table 2 come from an independent run that was not part of the five runs used to compute the mean and standard deviation in Table 1, which naturally leads to slight variability. All experiments use exactly the same configuration. To ensure consistency and maintain rigor, we have revised Table 1 by incorporating the results from Table 2 into the standard deviation calculation. We apologize for the confusion caused and have clarified this in the updated version.

---

> ### Author Response · Authors · 2025-11-27
>
> >Have you experimented with stronger pretrained teachers (e.g., CLIP-ViT) instead of ImageNet CNNs?
>
> Thank a lot for the question. Following the reviewer’s suggestion, we have added experiments using stronger pretrained teachers, including CLIP-ViT and the medical-domain BioMedCLIP. The results are now reported in Table 4 of the revised version.
>
> Table 4: Quantitative performance comparison of various backbone architectures employed as the teacher network across all datasets.
> ||CLIP|BiomedCLIP|
> |:------------------|:----:|:----------:|
> |**HIS 2-shot**|88.5|**89.2**|
> |**HIS 4-shot**|91.2|**92.1**|
> |**HIS 8-shot**|93.5|**94.3**|
> |**LAG 2-shot**|90.3|**91.5**|
> |**LAG 4-shot**|93.1|**95.2**|
> |**LAG 8-shot**|93.8|**95.7**|
> |**APTOS 2-shot**|97.8|**99.9**|
> |**APTOS 4-shot**|**99.9**|**99.9**|
> |**APTOS 8-shot**|99.9|**100.0**|
> |**RSNA 2-shot**|85.7|**91.2**|
> |**RSNA 4-shot**|90.4|**94.3**|
> |**RSNA 8-shot**|93.1|**96.2**|
> |**Brain Tumor 2-shot**|**92.8**|91.7|
> |**Brain Tumor 4-shot**|93.5|**94.6**|
> |**Brain Tumor 8-shot**|95.2|**98.1**|

---

### Author Response · Authors · 2025-12-03
**A Summary of Our Interactions with Reviewers**

We sincerely thank all reviewers, ACs, and SACs for their constructive feedback. Below is a concise summary of our interactions and how each concern was addressed.

***

`Reviewer gRTX`

**Main concerns:** (1) Accuracy of the term “distillation”, (2) fairness of baseline comparisons, (3) lack of pixel-level localization evaluation, (4) inconsistencies in table results, (5) under-exploration of stronger pre-trained teacher networks.

**Our response:**

(1) Provided explanation and justification for using the term “distillation” with supporting references (Section 1).

(2) Clarified that all baselines (MediCLIP, MVFA, INP-Former) were trained from scratch to ensure fair comparison (Section 4.1).

(3) Explained the focus on image-level detection, as many datasets in our diverse collection lack pixel-level annotations (Appendix E).

(4) Corrected tables for consistency (Table 1).

(5) Added experiments with additional teacher networks (CLIP-ViT and BioMedCLIP) (Table 4).

***

`Reviewer fFwa`

**Main concerns:** (1) Lack of pixel-level localization evaluation, (2) adaptation method for baselines, (3) selection of hyperparameter $\lambda$, (4) dataset scope for inference time and memory usage, (5) flexibility of the framework architecture, (6) absence of medical-domain pre-trained teachers.

**Our response:**

(1) Explained the focus on image-level detection because not all datasets in our diverse collection contain pixel-level annotations (Appendix E).

(2) Clarified that no modifications were made to methods such as PatchCore and RD4AD (Section 4.1).

(3) Provided details on the process for selecting $\lambda$ (Section 4.3).

(4) Clarified the dataset scope used for efficiency measurements (Section 4.5).

(5) Added explanation of the flexibility of the current architectural design.

(6) Included experiments with additional teacher networks (CLIP-ViT and BioMedCLIP) (Table 4).

***

`Reviewer 2kEb`

**Main concerns:** (1) Insufficient depth in analyzing the “learn-to-weight” mechanism, (2) impact of teacher network pre-training domain, (3) novelty formulation of the framework, (4) reasons for suboptimal performance of large networks (e.g., Swin), (5) details of the dataset “leave-one-out” protocol.

**Our response:**

(1) Added in-depth analysis of the “learn-to-weight” module, including comparisons of multiple similarity computation methods and visualizations (Appendix F).

(2) Conducted additional experiments with medical-domain pre-training (RadImageNet) (Appendix B).

(3) Clarified the framework’s innovative contribution in the revised manuscript (Section 1).

(4) Analyzed the potential “representation gap” in distilling Transformer features.

(5) Clarified that the “leave-one-out” protocol is applied at the dataset/task level.

***

`Reviewer 9qcV`

**Main concerns:** (1) Need to strengthen the introduction’s motivation, (2) inaccurate referencing, (3) insufficient informativeness of Figure 1, (4) misconception regarding “few-shot training.”

**Our response:**

(1) Revised and improved the relevant introduction paragraphs (Section 1).

(2) Clarified the argument supported by the cited reference.

(3) Updated Figure 1 to include feature space illustrations and loss computation details.

(4) Explicitly clarified the episodic training paradigm to avoid potential misunderstandings.

***

**All updates are highlighted in blue in the revised manuscript for easy reference.**

---

### Meta-Review · Area_Chair_pBs7 · 2025-12-15

**Summary:**

The work introduces a dual-distillation framework for few-shot anomaly detection in medical images, where a teacher-student distillation and a student self-distillation are jointly done. The approach is validated on five medical image AD datasets, in comparison to a large number of unsupervised and few-shot AD methods.

**Reviewer Concerns:**

The major concerns and how they are addressed are summarized in the following:

- Conceptual Clarity and Presentation

Concerns: Reviewers questioned the correctness of the term "distillation", the novelty and motivation of the framework, and the few-shot/episodic training.

AC: The authors clarified terminology and novelty well, strengthened the introduction, and explicitly explained the episodic training.


- Experimental Fairness and Baselines

Concerns: There were doubts about the fairness of baseline comparisons, the need for adapting baselines, and the limited exploration of strong or medical-domain teacher networks.

AC: The authors confirmed all baselines were trained from scratch without modification, and added experiments with stronger and medical-domain pre-trained teachers (such as CLIP-ViT, BioMedCLIP). These help clear the doubts on the experimental settings. One remaining concern the AC has is that the proposed method uses rather different network backbones from the competing methods, such as the CLIP-based ones.

- Evaluation Scope and Metrics

Concerns: Reviewers noted the absence of pixel-level AD evaluation and unclear dataset scope for efficiency measurements.

AC: The authors justified the focus on image-level detection due to missing pixel-level annotations on many datasets and clarified the datasets used for inference time and memory evaluation. The work can be further improved by evaluating pixel-level AD on those datasets that have anomaly segmentation masks.

- Methodological Depth and Analysis

Concerns: The learn-to-weight mechanism was insufficiently analyzed, hyperparameter selection lacked justification, framework flexibility was unclear, and the behavior of large models was unexplained.

AC: The AC is satisfied with the additional analyses, visualizations, and comparisons added during rebuttal. The authors have also elaborated on hyperparameter selection, architecture use, and the performance gaps in large Transformer models.

- Paper Title

One concern missing in the reviews is the paper title. The full paper is focused on medical image AD only, ignoring images from other domains such as industrial defects and natural images. The current title, which sounds like tackling a general image AD problem, mismatched the actual work. Terms like 'medical images' are required in the paper title.

**Reviewer Scores:**

There are four reviews, including two weak accepts and two weak rejects (Reviewers gRTX and 9qcV). Reviewer gRTX raised three major concerns, including those in the soundness of the technical design and empirical justification. The AC confirms that the rebuttal has satisfactorily addressed the concerns. The concerns from Reviewer 9qcV are all about clarity issues, which have also been properly clarified in the rebuttal and revision. Therefore, the AC recommends acceptance for this work.

---

### Decision · Program_Chairs · 2026-01-26

Accept (Poster)